# Semi-Empirical Models and Revision of Predicting Approaches of Tree Aboveground Biomass Assessments

Sacramento Corral-Rivas [1] , José Encarnación Luján-Soto [1], Tilo Gustavo Domínguez-Gómez [1],
José Javier Corral-Rivas [2] , Felipa de Jesús Rodríguez-Flores [3] , José-Guadalupe Colín [1],
José de Jesús Graciano-Luna [1] and José Návar [4],*

1 Tecnológico Nacional de México/Instituto Tecnológico de El Salto, Calle Tecnológico No 101, Colonia La Forestal, El Salto C.P. 34942, Mexico; sacra.corral@gmail.com (S.C.-R.); jelujan@gmail.com (J.E.L.-S.); gustavo_dguezg@hotmail.com (T.G.D.-G.); jose_colin@hotmail.com (J.-G.C.); gracluna@hotmail.com (J.d.J.G.-L.)
2 Facultad de Ciencias Forestales, Universidad Juárez del Estado de Durango, Boulevard del Guadiana 501, Ciudad Universitaria, Durango C.P. 34160, Mexico; jcorral@ujed.mx
3 Departamento de Tecnología Ambiental, Universidad Politécnica de Durango, Carretera Durango-Mexico Km Localidad Dolores Hidalgo, Durango C.P. 34300, Mexico; jesu_rgz@hotmail.com
4 Tecnológico Nacional de México/Instituto Tecnológico de Ciudad Victoria, Blvd Emilio Portes Gil No 1301 Pte., Victoria C.P. 87010, Mexico
* Correspondence: jose.navar@itvictoria.edu.mx

**Abstract:** Tree aboveground biomass (e.g., bole, branches, and foliage), $M$, plays key roles in forest management as it is the basis for evaluating the sink and flux of, for example, carbon and nitrogen, stand productivity, dendro-energy, litter & root biomass, hydrological parameters, among others. With the aim of further simplifying and understanding $M$, the central objective of this research was to review available techniques to develop, test, and validate two independent novel non-destructive, semi-empirical models using four major $M$ datasets: (i) the shape dimensional bio-physical, $M_{SD}$; and (ii) the restrictive mathematical, $M_{NR}$, models. The proposed models advance and test how each of both approaches: (i) constant or (ii) variable scalar coefficients perform when predicting $M$ with major assumptions bearing bio-physical principles. Results showed that $M$ has to be predicted eventually with variable scalar coefficients; both models predicted compatible $M$ figures; the evaluations matched the conventional equation well; and the independent data sets were well validated; the coefficients of determination, $r^2$, and the standard errors, Sx%, had values >96% and <20%, respectively, for most tested conifer tree species. In spite of demonstrating empirically and physically the ontogenetic-dependency of scalar coefficients, the $M_{NR}$ model, with constant $\beta$-scalar and variable $a$-intercept coefficients, performed slightly better, and precision appeared to be a function of the tree species growing in different forest ecosystems. Therefore, better parameterization advances for the testing and validation of the $M_{SD}$ model that uses variable scalar coefficients, which are consistent with ontogenetic principles, are preliminarily recommended for $M$ assessments. The updated revision of models, the independent development, the construction using different assumptions, the individual mathematical and bio-physical parameterization, the consistency on $M$ assessments, and the bearing of physical and biological properties are key pieces of scientific information presented in this report are required in modern forest management when predicting $M$ and associated variables and attributes.

**Keywords:** biomass allometry; $a$ and $B$ scalar coefficients; independent physical and biological parameterization; wood specific gravity; form factors; checking the consistency of biomass equations

## 1. Introduction

The central objective of this research was to review available predicting techniques to simplify $M$ assessments by developing, testing, and validating two independent non-destructive, semi-empirical models using comprehensive data sets. Tree aboveground

biomass is the living and dead matter in standing trees and includes foliage, branches, stumps, and timber. Tree $M$ assessments are currently the focus of intensive research due to its linkages with: (i) total and component primary productivity; (ii) sink and flux of several bio-geo-chemicals including carbon and nitrogen; (iii) hydro-climate regulation; (iv) alternate energy sources; (v) primary variable in several ecological models; (vi) biomass allocation factors; and (vii) other factors [1–6]. Tree $M$ is an important parameter in forest hydrology as well as it controls the major water storages and fluxes between forest compartments, soils, aquifers, rivers, and atmosphere [7]. For example, tree $M$ has been physically linked with: (i) the major input of litter that accumulates on the forest soil; and (ii) the allocation of resources to root systems that perforate the soil and add important biomass resources inside the forest soil. Forest and litter interception and rainfall redistribution in canopies, litter, and inside the soil are then in part a function of tree $M$ [8].

The development and application of allometric equations is the standard methodology for tree, stand, regional, and World $M$ assessments [6,9–11]. Tree $M$ equations are classified according to: (i) the spatial scale of interest into local, regional, national or World equations and (b) the parameter estimation method as empirical, semi-empirical and theoretical models [6]. These two kinds of model classifications can be applied at the scale of single species or complete forests. Many examples of allometric equations for single tree species had been developed and reported in the scientific literature, e.g., [12–14]. Typical allometric equations developed for complete forests had been also reported [1,11].

Compilations of $M$ equations [3,14–16] report the most common empirical allometric model is the logarithmic equation where $M$ is estimated as a log linear function of diameter at breast height, $D$, with the scaling coefficients $a$ and $B$ [17], called hereafter the $M_C$ model. The compilation of approximately 500 $M$ equations for Latin American forests classified further these empirical equations according to the statistical technique employed in parameter estimation methods into log-linear (82.6%), non-linear (12.0%), seemingly un-related linear (3.9%), and non-linear seemingly un-related (0.6%) regression analysis [6] Other non-destructive empirical methods of $M$ evaluations had been reported in the scientific literature [3,6].

A theoretical non-destructive technique using the fractals theory was derived [18], named hereafter the $M_{WBE}$ model. The main assumption of this model is that $D$ is related to $M$ by $M \propto D^{8/3}$, indicating that the scaling exponent $B_{WBE}$ equals $8/3 = 2.67$ weighted by a $C$ proportionality constant and the wood specific gravity of the entire tree, $\rho$ [19,20]. Details on how to evaluate the $C$ coefficient were proposed earlier [6]. Using taper approaches that describe the shape of tree boles, Návar [6] noted the $B_{WBE}$ scaling exponent may oscillate between 2.33 and 2.67 when timber tapering shifts from conical to cylindrical shapes, stressing the $B_{WBE}$ fit well the upper limits of timber shapes and the fractal technique needs further refinement. Tree boles are neither fully conical nor cylindrical in shape but a mixture of different shapes and hence semi-empirical methods appear at this time to be the route to better model $M$. However, semi-empirical models with worldwide application are scarce in the scientific literature and hence the need for further model development, testing, validation, and application.

Two issues arise when contrasting the $M_C$ and the $M_{WBE}$ models: (i) Is the $B$-scalar a constant or variable coefficient? (ii) Is the $a$-intercept a constant or variable coefficient? Most conventional equations report different $a$-intercept and $\beta$-scalar coefficients. In part this variability has been attributed to sample size, the size distribution of trees in the sample, among others. Zianis and Mencuccini [3], Návar [6] and Paz-Pellat et al. [21] modeled and Genet et al. [22] found evidence in *Fagus* plantations the $B$-scalar coefficient varies with age of trees. Návar [6] derived mathematically the $a$-intercept and found it varies as a function of the shape of the tree and the wood specific gravity, $\rho$, value. Assuming $\rho$ remains constant from young to mature trees, the shape of tree boles appears to control the $C$ coefficient as well as the $B$-scalar coefficient. However, further research is required to associate these coefficients to tree forms and, as a consequence, to improve understanding on tree allometry. Paz-Pellat et al. [21] recently advanced how to mathematically derive the

scalar coefficients using generalized tree allometric concepts that are in part consistent with those derived earlier [2,4,6,22] but the biological dependency of these coefficients has been quite elusive for some time.

The empirical destructive equation and the theoretical approaches have shed light into tree biomass allometry. However, several issues remain poorly understood: (a) Are the scalar coefficients *a* and *B* constant or variable figures? (b) Can semi-empirical methods be proposed, developed, tested, and validated for the assessments of tree *M*? (c) Can the evaluation of the *B* and *a*-scalar coefficients be simplified using bio-physically rather statistical approaches? (d) Do the proposed semi-empirical non-destructive models reproduce *M* assessments within the confidence bounds of the conventional equation? We addressed these questions by analyzing four compiled comprehensive World data sets of *M* equations by Ter Mikaelian and Korzukhin [15], Jenkins et al. [23] and Návar [12,16].

## 2. Materials and Methods

### 2.1. Tree Aboveground Biomass Allometric Models

Classic models that predict *M* reported in the scientific literature are: (i) the conventional logarithm equation with the exogenous variable *D* [17,24–26], (ii) the linear equation with the exogenous variable $D^2H$ [26], (iii) the power equation with *D* [26], (iv) multiple linear equation [26], (v) multiple non-linear equation [26], (vi) among others. The evaluation of scalar coefficients can be carried out in conventional log-linear, linear, non-linear, multiple linear and multiple non-linear regression techniques as well as on seemingly unrelated linear regression and seemingly unrelated non-linear regression [27–30]. Attempts to find or to construct more physically based models had been conducted and reported [2,3,6,18].

#### 2.1.1. The Conventional Allometric Model

The conventional empirical statistically parameterized allometric equation reported in most compilations and meta-analysis case studies [6] is presented in Equation (1):

$$\text{Ln}(M_C) = \text{Ln}(a) + \beta \text{Ln}(D) \pm e_i; \quad M_C = aD^\beta \pm e_i \tag{1}$$

where: *a* and *B* are the scalar intercept and exponent coefficients, respectively; $e_i$ = the error.

#### 2.1.2. The Fractal Model

The theoretical non-destructive model proposed by West et al. [18] is described by Equation (2):

$$M_{WBE} = C\rho D^{\frac{8}{3}} \tag{2}$$

where: *C* = a proportionality constant, and $\rho$ = the entire tree specific gravity value. The scalar exponent, $B_{WBE}$, is fixed to $8/3 = 2.67$ and wood specific gravity is referred as the total tree specific gravity value, a weighted average of wood, bark, branches and leaves.

Equations (1) and (2) bear the following common properties:

$$a = C\rho; \quad B_{WBE} \neq B; \quad B_{WBE} = 2.67.$$

The notion the *B*-scalar coefficient is a constant value has spurred recent research to facilitate the development of semi-empirical allometric models. Ketterings et al. [2] simplified the dimensionality of Equation (2) by proposing and assuming that *D* scales to $2.0H^*$; where $H^*$ is the slope value of the *H* = f(*D*) ($\approx \text{Ln}(H) = \text{Ln}(a) + \beta \text{Ln}(D)$) relationship, e.g., $D^{2.0H^*}$. This approach may have a final constant approximate value of 2.5 as $\beta \approx 0.50$, which appears to be a constant figure for the ($\approx \text{Ln}(H) = \text{Ln}(a) + \beta \text{Ln}(D)$) function. However, the $D^{2.0}$ exponent assumes tree boles have cylindrical shapes.

### 2.2. Bio-Physics of Volume and Mass of Tree Boles

2.2.1. Shapes of Tree Boles

Focusing exclusively on tree bole as it explains more than 80% of *M*, tree boles have in general a quasi-conic or a quasi-cylindrical shape although parts are classified as cylindrical, neiloids, frustrum, and conical; moving from the base to the tip of the tree. In general, when trees are young the bole has a quasi-conical shape and adds mass as they increment in diameter and height, as they reach the dominant position in the stand they slow growth in height, add mass preferentially in diameter all along the bole shifting relentlessly from conical to cylindrical shapes from *D* to the end of the clean stem just before branches protrude the stem, changing the shape from quasi-conical to quasi-cylindrical as they grow over time. Using taper functions, Návar et al. [13,31] presented several examples of these forms for several Mexican coniferous tree species. Then, it appears both scalar coefficients shift as trees grow over time because tree boles change in shape and the rate of biomass increases preferentially in diameter above *D* over time as trees slow growth in *H*.

2.2.2. Ontogenetic Principles

Using the tree bole geometric concepts described before; bearing in mind that boles are not two dimensional surfaces neither three dimensional photosynthetic domains; simulating a tree bole with *D* = 20 cm, *H* = 20 m, and $\rho$ = 0.50; as well as assuming tree is growing in different environments and it does not change dimensions and replicating the following shapes: (i) conical, (ii) first 5 m as cylinder and last 15 m as conical; (iii) half cylinder and second half cone, (iv) the first 15 m as cylinder and the last 5 m as cone, and (v) full cylinder; the following *B*-scalar coefficients assessed correctly *M* using the conventional Equation (1): (i) 2.230, (ii) 2.410; (iii) 2.485, (iv) 2.546, and (v) 2.598, respectively. Note that stocking either natural or using forest management practices, as well as tree diversity modify the shape of tree boles described before. Because the assembled average $C\rho = a = 0.13$ scalar intercept remains within the range of expected values in real tree boles, it is expected that the fractal dimension could be tuned by:

$$D^{\frac{7}{3}} \propto M \propto D^{\frac{8}{3}} \tag{3}$$

In the same way, modifying the *a*-scalar intercept to provide the coefficients 2.333 for conical as well as 2.667 for cylindrical shapes, the new *a*-intercept values would be: (i) 0.096 for conical and (v) 0.107 for cylindrical shapes of tree boles, respectively. However, for quasi-cylindrical shapes, when the *a*-intercept value remains constant, 0.107, the *B*-scalar coefficient has to be modified into (ii) 2.477, (iii) 2.553, and (iv) 2.614, respectively. These values exceed a bit the *B*-scalar coefficient average and standard deviation values described later (2.3699 $\pm$ 0.278) for nearly 300 *M* World case studies [3]. When leaving the *a*-scalar intercept value of the conical shape of tree boles as 0.096, the new *B*-scalar coefficient values would be for: (ii) 2.515, (iii) 2.590, and (iv) 2.651, respectively. These values exceed even further the *B*-scalar coefficient average and standard deviation values described above. New *a*-intercept values that fit the World average value (2.3699) would be: (i) 0.086, (ii) 0.147, (iii) 0.185, (iv) 0.221, and (v) 0.260, respectively. Then, it appears it is safe to assume that any theoretical model including the fractal one would have the following extreme *a*-scalar intercept values;

$$a = 0.08 \propto M \propto a = 0.26 \tag{4}$$

And as a consequence, when $\rho \approx 0.50$;

$$C = 0.16 \propto M \propto C = 0.52 \tag{5}$$

2.2.3. Further Empirical Evidence of Ontogenetie of Scalar Coefficient Values

Using all 41 *M* equations reported for North American tree species [15] the following simulations were conducted: (i) a variable *a*-scalar intercept value and a constant *B*-scalar coefficient value of 2.38; (ii) a variable *B*-scalar coefficient value and a constant *a*-scalar inter-

cept value of 0.11; (iii) a variable *a*-scalar intercept value and a constant *B*-scalar coefficient value of 2.67 of the $M_{WBE}$ model; and (iv) a variable *a*-scalar intercept value and a variable *B*-scalar coefficient value. Results of these simulations contrasted with the 41 equations for North American tree species are reported in Figure 1, as Figure 1a–d, respectively.

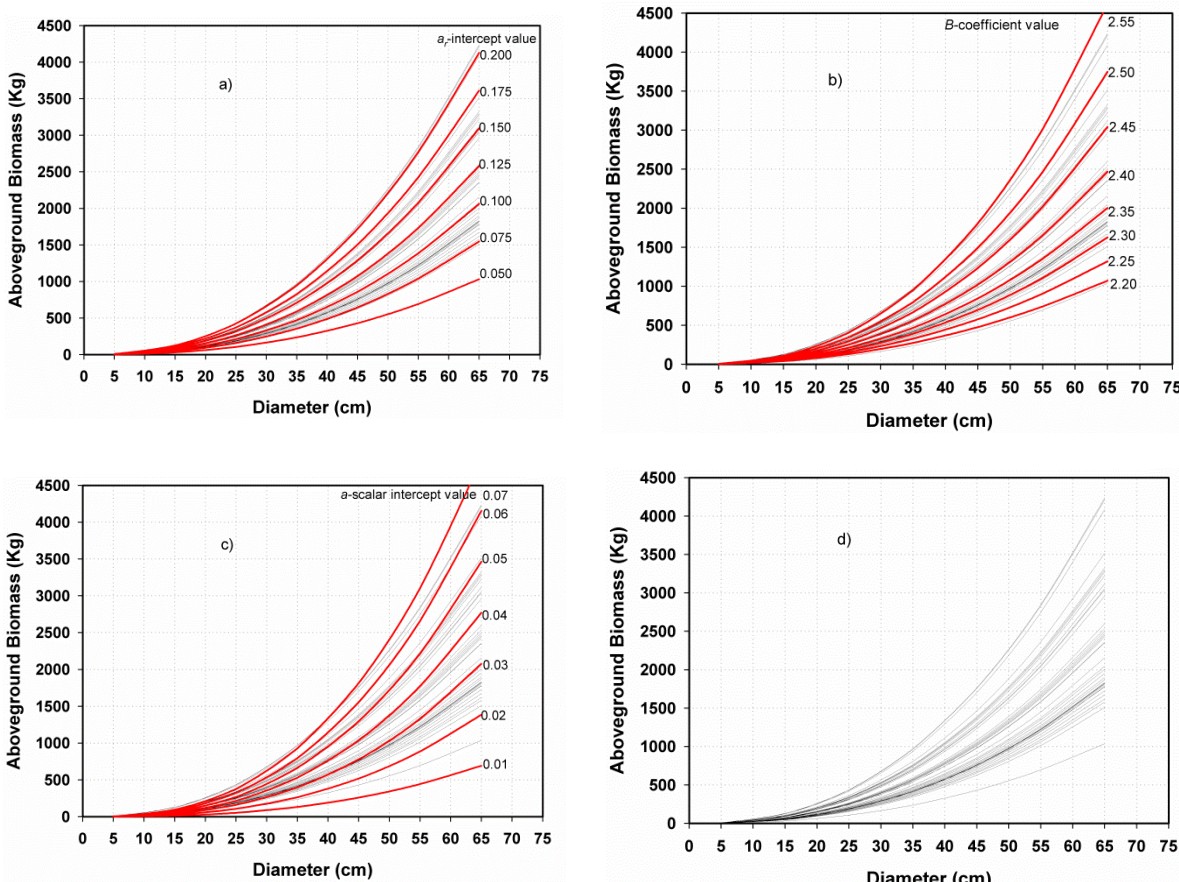

**Figure 1.** Tree aboveground biomass equations for all 41 North American tree species reported by Ter Mikaelian and Korzukhin [15] overlapped with allometric equations that assume: (**a**) a variable *a*-scalar intercept value and a constant *B*-scalar coefficient value of 2.38; (**b**) a constant *a*-scalar intercept value and a changing *B*-scalar coefficient value; (**c**) a variable *a*-scalar intercept value and a constant *B*-scalar coefficient value of the $M_{WBE}$ model of 2.67; and (**d**) a variable *a*-scalar intercept value and a variable *B*-scalar coefficient value. Red lines depict models with specific assumptions and grey lines the aboveground biomass equations reported by Ter Mikaelian and Korzukhin [15] for North American forests.

The red lines of Figure 1a match more consistently the empirical *M* dataset for all 41 North American tree species. The last Figure 1d represent the same empirical equations with varying coefficients of *a* and *B*. Note how Figure 1b,c deviate more notoriously from empirical equations pointing to the likelihood that for temperate and boreal tree species models with only *a*-scalar intercept varying values would be sufficient to evaluate *M* with a constant *B*-scalar coefficient a bit smaller than that of the $M_{WBE}$ model. However, from the geometric principles of tree growth discussed above, as tree boles move from conical to cylindrical shapes both the *a* and *B* scalar coefficients should more likely be modified; as the former is a shape and the second is an acceleration constant value that increases with tree growth and height slows; *D* is the only variable explaining the change of *M* with a change of *D*.

### 2.3. Proposed Semi-Empirical Non-Ddestructive Models of M Assessments

The descriptions of Figure 1a,d could be modeled using two semi-empirical approaches, one for each one, called hereafter: (i) the reduced model, $M_{NR}$, and (ii) the shape-dimensional model, $M_{SD}$. The $M_{NR}$ model assumes the constant $B$-scalar coefficient value can be found in World $M$ case studies and the varying $a$-scalar intercept value can be modeled as a function of the specific gravity value of wood as well using sample size as a weighting factor. The $M_{SD}$ model assumes the varying $B$-scalar coefficient value can be derived using bio-physical principles and can be used to derive the $a$-scalar intercept value as a preliminary approach in the meantime better data to test proposed bio-physical assessments of the $a$-intercept become available.

#### 2.3.1. The Reduced $M_{NR}$ Model

A reduced $M_{NR}$ model can be proposed and constructed by assuming: (a) it has a constant $B$-slope coefficient value that can be preliminarily approximated using meta-analysis case studies of World $M$ allometry; (b) the conventional bole wood specific gravity, $\rho_w$, is a good estimator for the entire tree specific gravity value ($\rho_w = \rho$); and (c) $\rho_w$ and the $a$-scalar intercept data are functionally related with a slope coefficient describing the $C$ proportionality constant of Equation (2). These assumptions fully and consistently meet the principles of the $M_{WBE}$ as well as the $M_C$ models. Then, the proposed reduced model in this report that is similar to the $M_{WBE}$ model but that is parameterized in a different manner than the conventional model is:

$$M_{NR} = (a_r D^{B_n}); \; a_r = f(\rho_w); \; C = \frac{a_r}{\rho_w}; \; M_{NR} = C\rho_w D^{\beta_n} \tag{6}$$

where $a_r$ is the re-escalated $a$-intercept; $D$ is the diameter at breast height; $\rho_w$ is the the bole wood specific gravity; and $Bn$ is the scaling power coefficient to be found in meta-analysis $M$ case studies.

Statistics of several compilations of meta-analysis $M$ case studies used for finding the approximate $B$-scalar coefficient value using a re-escalated $a$-scalar intercept figure are reported in Table 1.

**Table 1.** The allometric model scalar coefficient figures with a re-escalated $a$-scalar intercept values for six meta-analysis tree aboveground biomass case studies.

| | | $A$ | | | $a$-Re-Escalated | | | $B$ | | |
|---|---|---|---|---|---|---|---|---|---|---|
| | N | $\bar{x}$ | $\Sigma$ | CI | $\bar{x}$ | $\sigma$ | CI | $\bar{x}$ | $\sigma$ | CI |
| Jenkins et al. [23] | 10 (2456) | 0.11 | 0.03 | 0.02 | 0.12 | 0.03 | 0.02 | 2.40 | 0.07 | 0.05 |
| Ter Mikaelian and Korzukhin [15] | 41 | 0.15 | 0.08 | 0.03 | 0.11 | 0.04 | 0.01 | 2.33 | 0.17 | 0.05 |
| Fehrmann and Klein [32] | 28 | 0.17 | 0.16 | 0.06 | 0.12 | 0.02 | 0.01 | 2.40 | 0.25 | 0.09 |
| Návar [16] | 78 | 0.16 | 0.15 | 0.03 | 0.14 | 0.09 | 0.02 | 2.38 | 0.23 | 0.05 |
| Návar [12] | 34 | 0.10 | 0.11 | 0.04 | 0.12 | 0.05 | 0.02 | 2.42 | 0.25 | 0.08 |
| Zianis and Mencuccini [3] | 277 | 0.15 | 0.13 | 0.01 | 0.12 | 0.04 | 0.01 | 2.37 | 0.28 | 0.03 |
| Mean Values | | 0.14 | 0.11 | 0.03 | 0.12 | 0.05 | 0.01 | 2.38 | 0.21 | 0.06 |

N, number of biomass equations; $\bar{x}$, average coefficient value; $\sigma$, standard deviation; CI, confidence interval values ($\alpha$ = 0.05; D.F = n − 1); $\mu$, population mean. Jenkins et al. [23] compiled 2456 clustered in 10 biomass equations for temperate North American tree species. Ter Mikaelian and Korzukhin [15] reported equations for 67 North American tree species but only 41 equations were employed that reported for total aboveground biomass. Návar [16] reported a meta-analysis for 229 allometric equations for Latin American tree species but only 78 fitted the conventional model for aboveground biomass. Návar [6] reported $B$-scalar exponent values for 34 biomass equations calculated from shape-dimensional analysis. Zianis and Mencuccini [3] reported equations for 279 worldwide species. It is recognized that this meta-analysis study overlaps several equations that were employed by Jenkins et al. (2003); Zianis and Mencuccini [3] and by Ter Mikaelian and Korzukhin [15].

A mean (confidence interval) $Bn$-scalar slope value of 2.38 ($\pm$0.06) is appropriate for the $M_{NR}$ model (6), when $D$ is measured at breast height for most trees growing in temperate and boreal forests. It deviates from the $D^{2.0H*}$ [2] and $D^{2.67}$ [18] and it describes better $M$ allometry for tree boles that fit better conical than cylindrical shapes. That is, most sampled trees appear to be in young stages and conical shapes fit better the tapering of tree boles.

In order to reduce biased $M$ evaluations, the $a$-scalar intercept has to be re-escalated, $a_r$, since the average $Bn$-scalar slope approximates better to a population mean coefficient while most allometric equations are derived from sample data. For available allometric equations with conventional and $Bn$-scalar slope values, Equation (7) takes care of this step:

$$a_r = \frac{aD_m^B}{D_m^{2.38}}; \; a_r = aD^{B-2.38} \tag{7}$$

where: $a_r$ is the re-scalated $a$-scalar intercept; and $D_m$ is the maximum diameter at breast height recorded in the allometric case study or the forest inventory.

Allometric Data for Testing the $M_{NR}$ Model

The proposed $M_{NR}$ model was constructed with 41 tree $M$ equations reported earlier [15] for North American temperate and boreal tree species. The procedure was the following; first, the re-escalated $a_r$-scalar intercept was estimated with Equation (7) and second the tree $M_{NR}$ was evaluated with Equation (6). The plotted red lines to see the feasibility of the re-escalated $a_r$ and proposed constant $B = 2.38$ scalar coefficient values are depicted in Figure 1a.

Next $a_r$ was plotted as a function of $\rho_w$ and a linear equation fitted to this relationship (Figure 2). The equation was forced to start from the origin in such a way that $a_r = C\rho_w$ with the slope describing the $C$ proportionality constant, to be consistent with both Equations (1) and (2). The bole wood specific gravity data was collected from internet sources.

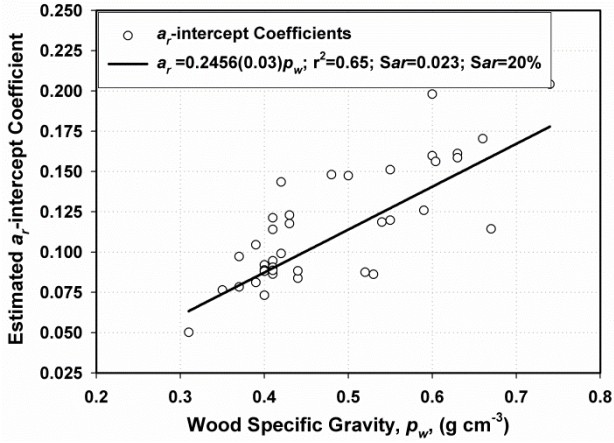

**Figure 2.** Relation of $a_r$-intercept and wood specific gravity coefficient values derived for 39 reported biomass equations for North American tree species. The regression line, equations, and goodness of fit parameters are also depicted.

Bole wood specific gravity data predicts well the derived $a_r$-re-escalated intercept coefficient value for all aboveground allometric equations, with the exception of two data points that were deleted from the regression analysis. Wood specific gravity values explained 65% ($\pm$2.3%) of the $a_r$-re-escalated intercept variance. Hence, the full semi-empirical preliminary $M_{NR}$ model for North American temperate and boreal tree species that can be applied for World temperate and boreal forests is described by model (8):

$$M_{NR} = \left(0.2457(\pm 0.015)\rho_w D^{2.38}\right) \tag{8}$$

The lower and upper confidence intervals (Sx = 0.095) of the slope coefficient for North American tree species are 0.2304 and 0.2609, respectively. Then, this reduced allometric equation entails only the $\rho_w$ parameter that is species specific to predict the $a_r$-scalar intercept and with a constant $Bn$ = 2.38 scalar value, tree $M$ of any temperate tree species can be easily derived.

#### Testing the $M_{NR}$ Model

The $M_{NR}$ model was tested with the 10 general equations developed and reported for USA [23]. In addition, 32 other equations collected from Mexico were used to validate and to contrast the restrictive as well as a second semi-empirical model named hereafter the shape-dimensional, $M_{SD}$, model.

#### 2.3.2. The Shape-Dimensional Model

A shape-dimensional model, $M_{SD}$, can be physically developed following the classic mass equation applied to tree boles in forestry. Tree $M$ is linearly and positively related to stem volume, $V$, and to wood specific gravity of the entire tree, $\rho$.

$$M = V\rho \tag{9}$$

A simple dimensional analysis of a perfect cylinder shows that the tree bole volume is

$$V = a_v D^2 H \tag{10}$$

where $a_v$ is 0.7854.

However, for real tree boles, the $a_v$ coefficient has been found to have an average value of 0.55 (for n = 50 case studies of conifer trees of northern Mexico) indicating that the stem cross-section at any height, or pieces of stems, has non-standard shapes that are only approximated by ideal objects. Fractal geometric analysis was developed to quantify the dimension of trees that cannot be described by Euclidean geometry [33]. It is generally acknowledged that a positive number between two and three is a better estimator of the tree's crown dimension, and it is assumed that the overall shape of a tree (stem and crown) may possess similar fractal dimensions. Then, in mathematical terms, the Schumacher and Hall [34] equation fits well bole volume:

$$V = a_v D^d H^h \tag{11}$$

where: $a_v \neq 0.7854$, $d$ and $h$ are positive numbers with $2 < d + h < 3$.

A statistical relationship between $H = f(D)$ ($\approx \mathrm{Ln}(H) = \mathrm{Ln}(a) + \beta \mathrm{Ln}(D)$) that convey biomechanical principles has been developed for forestry and botany studies [35–38] when stress-similarity for self-loading dictates the mechanical design of a tree, the power coefficient of this relationship, $\beta = H^*$, scale to $\approx$ ½ power of $D$ and a final constant $H^*$ is reached in old trees that reflects an evolutionary balance between the costs and benefits of stature [37]. In fact, Niklas [38] using meta-analysis case studies reported that $H^*$ approximates to 0.535 for a wide range of plant sizes. However, when using tree samples in a forest, this relationship is a function of sample size, tree species, stocking, site productivity, among other factors that may deviate the scaling factor from ½. Therefore, by assuming that;

$$H = \mathrm{Ln}(a) + \beta \mathrm{Ln}(D); \ \beta = H*; \ H = a_h D^{H*} \tag{12}$$

where $a_h$ is the intercept and $\beta = H^*$ is the slope of Equation (9), with $0 < H^* \leq 1; H^* \approx$ ½.

Then the Schumacher and Hall [34] Equation (11) can be simplified by using predicted $H$ by Equation (12). Hence Equation (11) becomes:

$$V = f\left(a_v D^d a_h D^{hH*}\right) = (a_v a_h) D^{d+hH*} \tag{13}$$

where: $a_v$ and $a_h$ = intercept coefficients that describe average taper and slenderness form factors of tree boles, respectively; $d$ and $h$ = power coefficients of the Schumacher and Hall Equation (11) [34]; $H*$ = power slope coefficient of the logarithmic $H$ vs. $D$ relationship.

Since tree biomass is assumed to be proportional to $V$ (with the tree wood specific gravity value as a weighting factor of the shape coefficient), then $M = f(D^{d+HB*})$ and in conjunction with Equation (13):

$$B_{theoric} = d + hH* \tag{14}$$

And,

$$a_{theoric} = (a_v a_h)\rho; \ a_{theoric} = C\rho; \ C = (a_v a_h) \tag{15}$$

Equation (14) was proposed earlier [2,3] as an aid to test the hypothesis that the scaling coefficient between $M$ and $D$ depends on the scaling coefficient between $H$ and $D$. Once these authors somehow proved correct this hypothesis, the $H*$ coefficient of the $H$-$D$ relationship was further used in empirically calculating the $B$-scalar exponent of Equation (1). Using this data set, the equation: $B = 1.9262 + 0.6972H*$; $r^2 = 0.42$ was developed. Please note that when $H* \approx 0.535$; $B \approx 2.30$, whose value approaches the lower limit of the conical shape of tree boles (2.33) stressing this approach eventually bias $M$ as trees grows over time and timber shape deviates further from conical forms.

Finally, the fully theoretical $M_{SD}$ model that requires the following relationships $V = f(D, H)$ and $H = f(D)$, in addition to $\rho$ is;

$$M_{SD} = \rho(a_v a_h) D^{d+hH*} \tag{16}$$

Model (16) can be simplified by further assuming $\rho = \rho_w$. Burkhardt and Tomé [36] and Genet et al. [22] proposed the taper form factor can be simplified as it is equivalent to the $a_v$ coefficient of the Schumacher and Hall equation, as follows:

$$F = \frac{v}{D^2 H} \cong a_v \tag{17}$$

Návar [6] proposed the coventional slenderness factor can be preliminarily approximated with the use of Equation (18) and is equivalent to the $a_h$ coefficient of the Schumacher and Hall equation, as folows,

$$N = \frac{H}{D} \cong a_h \tag{18}$$

where $v$ is the bole volume (m³); $D$ is the diameter at breast height in m; and $H$ is the top height (m).

Meta-analysis of $M$ case studies noted that the scalar coefficients $a$ and $B$ are related to one another in a negatively power function because (i) they seem to compensate the general principles of $M$ and (ii) the remaining variation must be partially explained by high values of both $a$ and $B$-scalar factors that would often result in large values of $M$ for large diameters that possibly approach the safety limits imposed by mechanical self loading [3,5,14,16]. All of the power equations reported by these authors project compatible $a$-scalar intercept values when $B \geq 2.33$. Using these statements, the $M_{SD}$ model (16) can be transformed into an hybrid physical-statistical model in the absence of $\rho$ or $\rho_w$ data and precise taper, $a_v$, and slenderness, $a_h$, coefficient values, the $a$-intercept coefficient can be preliminary statistically derived as a function of $B_{theoric}$, as follows;

$$a_{theoric} = (a_v a_h)\rho = f(B_{theoric} = d + hH*) \tag{19}$$

With these statements, the second version of the $M_{SD}$ semi-empirical model that is physical-statistically parameterized is;

$$M_{SD} = a_{theoric} = (a_v a_h)\rho = f(B_{theoric} = d + hH*) \tag{20}$$

The $a_h$ and $a_v$ coefficients are hard to be precisely defined when most inventoried trees meet the $D \geq 7$ cm dimensions providing intercept values with large variances, then the theoretical semi-empirical $M_{SD}$ model (16) and the physical-statistical $M_{SD}$ model (19) become in part semi-empirical models. Using Zianis and Mencuccini [3] World $M$ data set, the fully statistically parameterized equation is: $M_{SD} = [(7.0281 * B^{-4.7558}) * D^{(1.9262 + 0.6972H*)}]$; where the equation within the first bracket calculates the varying $a$-scalar intercept and the equation within the second bracket estimates the varying $B$-scalar exponent of the conventional $M_C$ equation. This empirical model assumes $B = B_{theoric}$ for any tree species. The $M_{SD}$ model in two forms Equations (16) and (19) advance the statistical model proposed earlier as they develop further into the calculation of varying $a$ and $B$-scalar coefficients instead of exploring simple empirical equations that calculates and associates the $B$-scalar coefficient as a function of only the $H$-$D$ relationship.

The $M_{SD}$ model when transformed into a physical-statistical technique has several major disadvantages: (a) the inherent co-linearity problems of estimating $a$ with $B_{theoric}$, (b) the log-relationships between $V = f(D, H)$ and $D = f(H)$ are required in order to estimate $a$ and $B_{theoric}$ figures, (c) measurements or data of $\rho$ is required, and (d) an empirical equation that relates $a$ to $B_{theoric}$ has to be developed on site or alternatively use preliminary equations previously reported [3,5,14] or the one developed for this report. However, all of these equations would consistently evaluate compatible average $a$-intercept values with an estimated $B_{theoric}$-slope coefficient.

Allometric Data for Testing the $M_{SD}$ Model

Two independent $M$ datasets tested and validated set hypotheses of the $M_{SD}$ model. A list of biomass, bole volume, and $H$-$D$ allometric equations including statistical parameters was compiled. The first data set included 21 $M$ case studies that fitted the $M_{NR}$ as well as the $M_{SD}$ models to estimate the $B_{theoric}$ and $a_{empiric}$ values and to develop an empirical equation to estimate $a$ (named $a_{theoric}$ because it results from $B_{theoric}$ values). This data set was statistically analyzed to provide site-specific-species $M$ equations, using most of the biomass data already published by Návar [14]. This data source was previously used to develop biomass equations for each individual tree species at the regional scale, within a linear radius of 500 km in the Sierra Madre Occidental Mountain Range of Durango and Chihuahua, Mexico. The second independent data source makes up 11 $M$ case studies that were collected to have an independent data set and to validate the proposed $M_{SD}$ as well as the $M_{NR}$ models.

The rationale behind was that different on and offsite allometric models predict different $M$ values, and therefore they exhibit large variation for similar forest ecosystems, as was seen in studies for tropical trees [39–43]; tropical dry trees ([14]; as well as for the IAN 710 hybrid of *Hevea brasiliensis* trees ([44–46]. Therefore, any semi-empirical model must also evaluate tree $M$ that fall within the variation of all of these biomass equations or within the variation of a single allometric equation.

Allometric Data for Validating the $M_{SD}$ Model

A total of 32 $M$ case studies were collected; 21 for testing and 11 for validating the $M_{SD}$ model. The following information was recorded: the tree species and the region for which the equation was developed, the $a$ and $B$ values, the coefficient of determination $R^2$, the mean square error, the MSE, the standard error for $B$ ($S_B$), and the $D$ range of harvested trees (Table 2). For all 32 data sets, 11 broadleaved species belonging to five genera were studied, while eight coniferous tree species from *Pinus* spp. were recorded. The species growing in tropical dry regions are generally referred to as tropical dry trees, and a single allometric equation is reported for five of these species. In all of the 32 case studies, the $D$

interval of sampled trees was in the range of 1.4 to 62.5 cm, with a mean of 20.0 cm. The collected information is assumed to represent a sample of the *M–D-H* relationships for tree species growing in the northern forests of Mexico, but also including one species from southeastern Mexico.

**Table 2.** Parameters of the conventional model (1) employed in the development and testing performance of the semi-empirical non-destructive models for forest ecosystems of northern Mexico.

| Location | Species (n) | Coefficient Values of Equation (1) | | | | | Diameter Statistics (cm) | | |
|---|---|---|---|---|---|---|---|---|---|
| | | $a$ | $B$ | $S_B$ | $r^2$ | MSE | Min | Max | Mean |
| DATA FOR MODEL DEVELOPMENT (n = 21) | | | | | | | | | |
| 1. S. Chihuahua | *P. arizonica* (n = 30) | −1.482 | 2.129 | 0.1697 | 0.84 | 0.026 | 16.20 | 32.90 | 25.70 |
| 2. S. Chihuahua | *P. durangensis* (n = 30) | −3.532 | 2.731 | 0.1478 | 0.92 | 0.054 | 12.10 | 46.00 | 27.40 |
| 3. S. Chihuahua | *Qurcus* spp. (n = 45) | −2.144 | 2.403 | 0.1275 | 0.89 | 0.060 | 15.40 | 48.70 | 29.10 |
| 4. El Salto, Dgo | *P. cooperi* (n = 20) | −1.922 | 2.321 | 0.1596 | 0.93 | 0.068 | 12.50 | 57.40 | 31.70 |
| 5. El Salto, Dgo | *Q. sideroxylla* (n = 30) | −2.592 | 2.585 | 0.1093 | 0.95 | 0.061 | 9.80 | 62.50 | 27.80 |
| 6. Tepehuanes, Dgo | *P. arizonica* (n = 36) | −3.573 | 2.746 | 0.0897 | 0.96 | 0.038 | 10.00 | 45.00 | 22.60 |
| 7. Tepehuanes, Dgo | *P. durangensis* (n = 15) | −3.416 | 2.715 | 0.1405 | 0.96 | 0.039 | 11.80 | 57.20 | 24.30 |
| 8. Tepehuanes, Dgo | *P. leiophylla* (n = 12) | −3.039 | 2.523 | 0.2237 | 0.92 | 0.058 | 13.90 | 34.80 | 21.30 |
| 9. Altares, Dgo | *P. arizonica* (n = 60) | −0.877 | 1.980 | 0.0560 | 0.81 | 0.094 | 9.90 | 45.00 | 25.70 |
| 10. San Dimas, Dgo | *P. ayacahuite* (45) | −3.066 | 2.646 | 0.0690 | 0.97 | 0.044 | 5.70 | 30.30 | 15.40 |
| 11. San Dimas, Dgo | *P. cooperi* (n = 12) | −3.264 | 2.707 | 0.1100 | 0.90 | 0.274 | 8.20 | 38.10 | 18.40 |
| 12. San Dimas, Dgo | *P. durangensis* (n = 71) | −2.084 | 2.323 | 0.0680 | 0.94 | 0.074 | 6.20 | 48.50 | 18.70 |
| 13. San Dimas, Dgo | *P. leiophylla* (n = 15) | −3.549 | 2.787 | 0.1020 | 0.94 | 0.065 | 9.60 | 29.00 | 20.20 |
| 14. Mezquital, Dgo | *P. oocarpa* (31) | −3.065 | 2.625 | 0.1030 | 0.93 | 0.061 | 12.20 | 44.80 | 25.20 |
| 15. Mezquital, Dgo | *P. pseudostrobus* (n = 24) | −2.611 | 2.531 | 0.2700 | 0.88 | 0.047 | 12.00 | 32.00 | 19.60 |
| 16. Mezquital, Dgo | *P. teocote* (n = 49) | −3.182 | 2.702 | 0.0690 | 0.96 | 0.050 | 7.30 | 43.30 | 21.90 |
| 17. Mezquital, Dgo | *Quercus* spp. (n = 17) | −2.754 | 2.574 | 0.0700 | 0.94 | 0.089 | 7.30 | 41.20 | 21.10 |
| 18. Topia, Dgo | *P. durangensis* (n = 60) | −2.108 | 2.373 | 0.0606 | 0.96 | 0.019 | 11.80 | 48.40 | 26.00 |
| 19. E. Sinaloa | *Tropical Dry trees* (n = 40) | −2.523 | 2.437 | 0.1993 | 0.80 | 0.443 | 5.20 | 32.60 | 14.80 |
| 20. Dgo.-S.Chih. | *Pinus* spp. (n = 520) | −2.818 | 2.574 | 0.0260 | 0.94 | 0.076 | 5.70 | 57.40 | 23.50 |
| 21. Dgo.-S.Chih. | *Quercus* spp. (n = 106) | −2.874 | 2.631 | 0.0807 | 0.93 | 0.078 | 7.30 | 62.50 | 26.80 |
| Average | | −2.69 | 2.53 | 0.12 | 0.92 | 0.09 | 10.00 | 44.65 | 23.20 |
| C.I. | | 0.30 | 0.09 | 0.03 | 0.02 | 0.04 | 1.38 | 4.50 | 1.89 |
| DATA FOR MODEL MODEL VALIDATION (n = 11) | | | | | | | | | |
| 22. Iturbide, N.L. | *P. pseudostrobus* (n = 8) | −3.164 | 2.599 | **0.0807** | 0.98 | NA | 5.00 | 42.40 | 2.32 |
| 23. Iturbide, N.L. | *Q. cambyi* (n = 8) | −2.311 | 2.449 | **0.0807** | 0.97 | NA | 5.00 | 39.50 | 23.10 |
| 24. Iturbide, N.L. | *Q. laceyi* (n = 7) | −2.434 | 2.507 | **0.0807** | 0.98 | NA | 6.00 | 35.20 | 20.20 |
| 25. Iturbide, N.L. | *Q. risophylla* (n = 8) | −2.209 | 2.374 | **0.0807** | 0.97 | NA | 7.40 | 40.60 | 23.90 |
| 26. El Salto, Dgo | Y. Pine trees N.L. (n = 17) | −0.610 | 1.713 | 0.1073 | 0.94 | 0.068 | 1.80 | 14.60 | 7.70 |
| 27. El Salto, Dgo | Y. *P. durangensis* (n = 25) | −3.642 | 2.746 | 0.2370 | 0.85 | 0.168 | 4.00 | 15.00 | 10.00 |
| 28. El Salto, Dgo | Y. *P. cooperi* (n = 19) | −3.119 | 2.588 | 0.1515 | 0.94 | 0.047 | 5.00 | 14.40 | 9.20 |
| 29. El Salto, Dgo | Y. *Pinus* spp. (n = 12) | −2.397 | 2.364 | 0.4276 | 0.73 | 0.281 | 3.80 | 13.60 | 9.70 |
| 30. NE Mexico | *Acacia* spp. (n = 190) | −1.414 | 2.114 | 0.0360 | 0.95 | 0.144 | 1.40 | 57.30 | 8.30 |
| 31. NE Mexico | *Prosopis* spp. (n = 62) | −1.871 | 2.320 | 0.0880 | 0.92 | 0.213 | 2.70 | 21.80 | 9.10 |

**Table 2.** *Cont.*

| Location | Species (n) | Coefficient Values of Equation (1) | | | | | Diameter Statistics (cm) | | |
|---|---|---|---|---|---|---|---|---|---|
| | | *a* | *B* | $S_B$ | $r^2$ | MSE | Min | Max | Mean |
| 32. Ver., Mexico | *Hevea brasiliensis* (n = 20) | −2.199 | 2.404 | 0.3246 | 0.74 | 0.152 | 20.00 | 50.00 | 31.10 |
| Average | | −2.31 | 2.38 | 0.15 | 0.91 | 0.15 | 5.65 | 31.31 | 14.06 |
| C.I. | | 0.50 | 0.16 | 0.07 | 0.05 | 0.05 | 3.00 | 9.44 | 5.29 |

Dgo., Durango; Chih., Chihuahua; Y, young; NA, not available; $S_B$, standard error for *B*; $r^2$, coefficient of determination; MSE, mean square error; Min, minimum; Max, maximum. Bold numbers means that data was derived from other regional studies.

For *M* case studies with different allometric equations, first, the scalar coefficients *a* and *B* were calculated by least square techniques in linear regression using the natural log transformation of *M* and *D*. Second, for all biomass databases, the *H-D* and *V-H,D* data were extracted from the same biomass studies mentioned previously. When this information was not available it was collected from several reports for the same species (Table 3). The statistical logarithmic relations fitted for calibrating the non-destructive model are described further below. The *H-D* and *V-H,D* relationships evaluated the $B_{theoric}$ coefficient by fitting Equation (14). The $a_{theoric}$ was calculated as a function of the $B_{theoric}$ coefficient by regressing these variables using the 21 tree *M* case studies.

**Table 3.** Diameter—height and volume—diameter— height relationships for model development and testing of the non destructive method of total aboveground biomass estimation.

| D-H Relationship; Ln(H) = a + BLn(D); H = Bo$D^{B*}$ | | | | V-D,H Relation; Ln(V) = a + BLn(D) + B1Ln(H); V = Bo$D^{B1}H^{B2}$ | | | | |
|---|---|---|---|---|---|---|---|---|
| (Equation No) $a_h$(Bo) | B = H* | $r^2$ | MSE | Ln ($a_v$(Bo)) | d(B1) | h(B2) | $r^2$ | MSE |
| EQUATIONS FOR MODEL DEVELOPMENT (n = 21) | | | | | | | | |
| (1) 1.6270 | 0.3040 | 0.22 | 0.0090 | −8.8600 | 1.8300 | 0.7700 | 0.92 | 0.0115 |
| (2) 0.7560 | 0.5440 | 0.43 | 0.0310 | −9.6600 | 2.0100 | 0.8600 | 0.98 | 0.0120 |
| (3) 0.5070 | 0.5630 | 0.34 | 0.0490 | −9.4300 | 1.9800 | 0.7200 | 0.98 | 0.0108 |
| (4) 1.0490 | 0.5620 | 0.73 | 0.0160 | −9.3200 | 1.8700 | 0.9360 | 0.96 | 0.0320 |
| (5) 0.7520 | 0.5650 | 0.62 | 0.0330 | −9.6700 | 1.7900 | 1.0270 | 0.98 | 0.0180 |
| (6) 0.4540 | 0.6780 | 0.47 | 0.0680 | −9.5600 | 1.8560 | 0.9990 | 0.99 | 0.0077 |
| (7) −0.0020 | 0.8420 | 0.75 | 0.0330 | −9.7600 | 2.0470 | 0.8570 | 0.99 | 0.0057 |
| (8) 0.7530 | 0.5690 | 0.69 | 0.0150 | −10.0900 | 2.2300 | 0.7530 | 0.99 | 0.0085 |
| (9) 1.7280 | 0.3230 | 0.33 | 0.0205 | −9.5370 | 1.9678 | 0.8590 | 0.99 | 0.0047 |
| (10) −0.0470 | 0.9070 | 0.87 | 0.0250 | −9.6370 | 1.6870 | 1.1710 | 0.99 | 0.0087 |
| (11) 0.6630 | 0.6260 | 0.89 | 0.0160 | −9.8870 | 2.0960 | 0.8770 | 0.99 | 0.0057 |
| (12) 1.3090 | 0.4730 | 0.62 | 0.0330 | −9.6960 | 1.9280 | 0.9560 | 0.99 | 0.0160 |
| (13) 0.7780 | 0.5980 | 0.75 | 0.0150 | −9.7920 | 2.1790 | 0.6940 | 0.99 | 0.0061 |
| (14) 0.7740 | 0.5750 | 0.47 | 0.0400 | −9.8440 | 1.9890 | 0.9350 | 0.99 | 0.0047 |
| (15) 0.9600 | 0.5950 | 0.75 | 0.0070 | −9.9590 | 1.6930 | 1.2910 | 0.99 | 0.0028 |
| (16) 0.9630 | 0.5360 | 0.66 | 0.0240 | −9.6320 | 2.0510 | 0.7890 | 0.99 | 0.0103 |
| (17) 1.2430 | 0.4410 | 0.53 | 0.0390 | −9.5500 | 1.8230 | 0.9760 | 0.98 | 0.0209 |
| (18) 1.2150 | 0.4660 | 0.46 | 0.0240 | −9.2170 | 1.9200 | 0.8040 | 0.99 | 0.0067 |

**Table 3.** *Cont.*

| (Equation No) $a_h$(Bo) | $B = H^*$ | $r^2$ | MSE | Ln ($a_v$(Bo)) | d(B1) | h(B2) | $r^2$ | MSE |
|---|---|---|---|---|---|---|---|---|
| **D-H Relationship; Ln(*H*) = *a* + BLn(*D*); *H* = Bo*D*$^{B*}$** | | | | **V-D,H Relation; Ln(*V*) = *a* + BLn(*D*) + B1Ln(*H*); *V* = Bo*D*$^{B1}$*H*$^{B2}$** | | | | |
| (19) 0.5717 | 0.3750 | 0.14 | 0.2130 | −9.7344 | 2.0163 | 0.8150 | 0.98 | 0.1726 |
| (20) 1.3392 | 0.4430 | 0.25 | 0.1010 | −10.0750 | 1.7284 | 1.3466 | 0.98 | 0.0230 |
| (21) 1.1537 | 0.4119 | 0.29 | 0.0610 | −9.3906 | 1.8961 | 0.8129 | 0.98 | 0.0160 |
| Average | 0.54 | 0.54 | 0.04 | −9.63 | 1.93 | 0.92 | 0.98 | 0.02 |
| C.I. | 0.06 | 0.09 | 0.02 | 0.12 | 0.06 | 0.07 | 0.01 | 0.02 |
| EQUATIONS FOR MODEL VALIDATION (n = 11) | | | | | | | | |
| (22) 0.9890 | 0.5540 | 0.60 | 0.0350 | −9.7105 | 2.0068 | 0.8562 | 0.99 | 0.0160 |
| (23) 1.1187 | 0.4209 | 0.79 | 0.0480 | −10.1246 | 1.6824 | 1.2601 | 0.91 | NA |
| (24) 1.2810 | 0.3330 | 0.78 | 0.0320 | −9.5341 | 1.5731 | 1.0773 | 0.86 | NA |
| (25) 0.9560 | 0.5090 | 0.89 | 0.0320 | −10.7150 | 1.7917 | 1.4429 | 0.95 | NA |
| (26) 1.9633 | 0.3842 | 0.91 | 0.0200 | −7.8209 | 2.5891 | −0.9378 | 0.96 | 0.0500 |
| (27) 1.2347 | 0.4750 | 0.50 | 0.0400 | −9.6237 | 2.0354 | 0.7972 | 0.99 | 0.0150 |
| (28) 0.6372 | 0.6680 | 0.80 | 0.0230 | −9.6414 | 1.8232 | 1.0820 | 0.99 | 0.0190 |
| (29) 0.9770 | 0.5530 | 0.57 | 0.0380 | −9.7418 | 2.0047 | 0.8757 | 0.99 | 0.0149 |
| (30) −0.4290 | 0.6180 | 0.45 | 0.0820 | −9.4756 | 1.9902 | 1.1737 | 0.99 | 0.0066 |
| (31) −0.4732 | 0.6272 | 0.57 | 0.0715 | −9.4756 | 1.9902 | 1.1737 | 0.98 | 0.0115 |
| (32) 0.5430 | 0.7450 | 0.63 | 0.0255 | −9.8310 | 1.8370 | 1.0010 | 0.93 | 0.0430 |
| Average | 0.54 | 0.68 | 0.04 | −9.61 | 1.94 | 0.89 | 0.96 | 0.020 |
| C.I. | 0.07 | 0.09 | 0.01 | 0.41 | 0.16 | 0.38 | 0.03 | 0.010 |

In order of causing no further confusion, *M* was estimated with the following two options: (a) the conventional model (1), $M_C$, with statistically evaluated $B_{empiric}$-$a_{empiric}$ and (b) the shape dimensional model, $M_{SD}$, that evaluated $B_{theoric}$-$a_{theoric}$. The $M_C$ Equation (1) confidence bands given by the standard error of $B_{empiric}$ ($S_B$), were calculated to find out whether *M* assessments from the $M_{SD}$ as well as from the $M_{NR}$ model would fall within this interval for a single equation and to note whether these models provide unbiased estimates that oscillates up and down the average line depicted by Equation (1).

### 2.3.3. Model Fitting Statistics

Measured tree *M* data is the raw biomass ($M_R$), tree *M* estimated by the conventional equation is $M_C$, and tree *M* evaluated by the shape dimensional, $M_{SD}$, and restrictive models is $M_{NR}$ for the 21 biomass data sets for fitting the parameters collected for the non-destructive model and for the 11 biomass data sets used for testing the field applicability of the proposed model for biomass estimates. The coefficient of determination, $r^2$, and the standard error as a percentage, Sx%, were evaluated by contrasting only $M_C$ vs. $M_{SD}$ and $M_C$ vs. $M_{NR}$; as well as $M_{SD}$ vs. $M_{NR}$ models.

### 3. Results

#### 3.1. The $M_{NR}$ Model

The proposed $M_{NR}$ model consistently matches reported allometric equations for all three case studies. In addition, the mean (confidence interval) $a_r$ values (0.12 ± 0.01) are statistically similar to the reported conventional mean *a*-scalar (0.14 ± 0.03) intercept coefficient figures (Table 1). It reproduces compatible *M* assessments as the standard error

was <20% for 81%, 60% and 51% of the allometric equations reported earlier [12,14,15,23] (Figure 3).

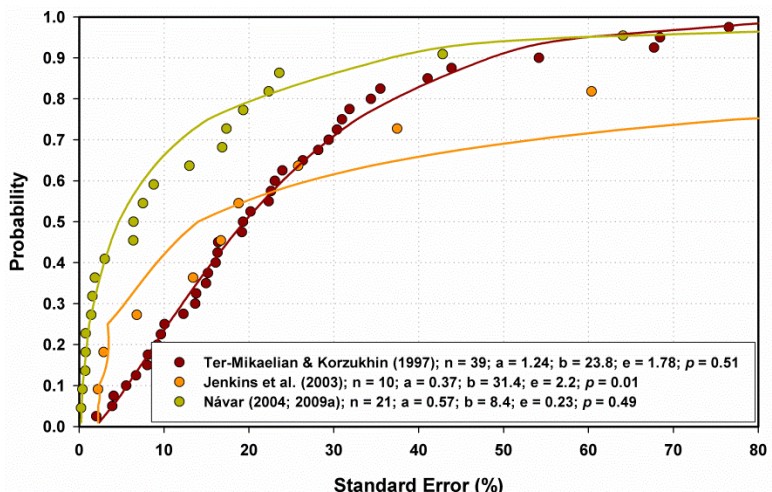

**Figure 3.** The probability of the standard error calculated from the difference of the conventional and proposed restrictive allometric model for three major aboveground biomass datasets. Lines represent the Weibull probability density function [12,15,16,23].

That is, only in 4 out of 21 (19%); in 4 out of 10 (40%); and in 19 out of 39 (49%); the standard error is larger than 20% for the compiled $M$ case studies, respectively. For all Návar's [12,14] 21 allometric equations, the standard deviation values > 40% could be observed for only the species *Pinus arizonica* and *Acacia* spp. but most of the error was found in trees with $D$ > 30 cm. An example of the application of the $M_{NR}$ model is depicted in Figure 4 in conjunction with the Jenkin's et al. [23] $M$ equations for ten clusters of North American tree species. For all ten Jenkins' et al. [23] equations standard deviation figures > 37% were observed in cedar/larch; true fir/hemlock; and woodland (juniper/oak/mesquite) species. The first two clusters of species recorded most deviations in trees with $D$ > 60 cm. The $M_{NR}$ model bias projected $M$ data for the woodland tree species. For the Ter-Mikaelian and Korzukhin's [15] 39 biomass equations, the proposed model deviates (e.g., Sx(%) < 30%) in 29 tree species.

Deviations of $M$ assessments present one of the smallest errors (15%) ever recorded in $M$ allometry since mean (confidence interval) standard errors values are: 23.9 (5.7); 20.4 (11.6) and 12.3 (6.6) for biomass equations had been previously reported [12–15,23]. For the studied tree species, the model reproduces unbiased $M$ estimates warranted by the regression line between the $a_r$-intercept and the $\rho_w$ data source as well as by the random oscillation of the proposed $M_{NR}$ model for the $M$ case studies tested and validated.

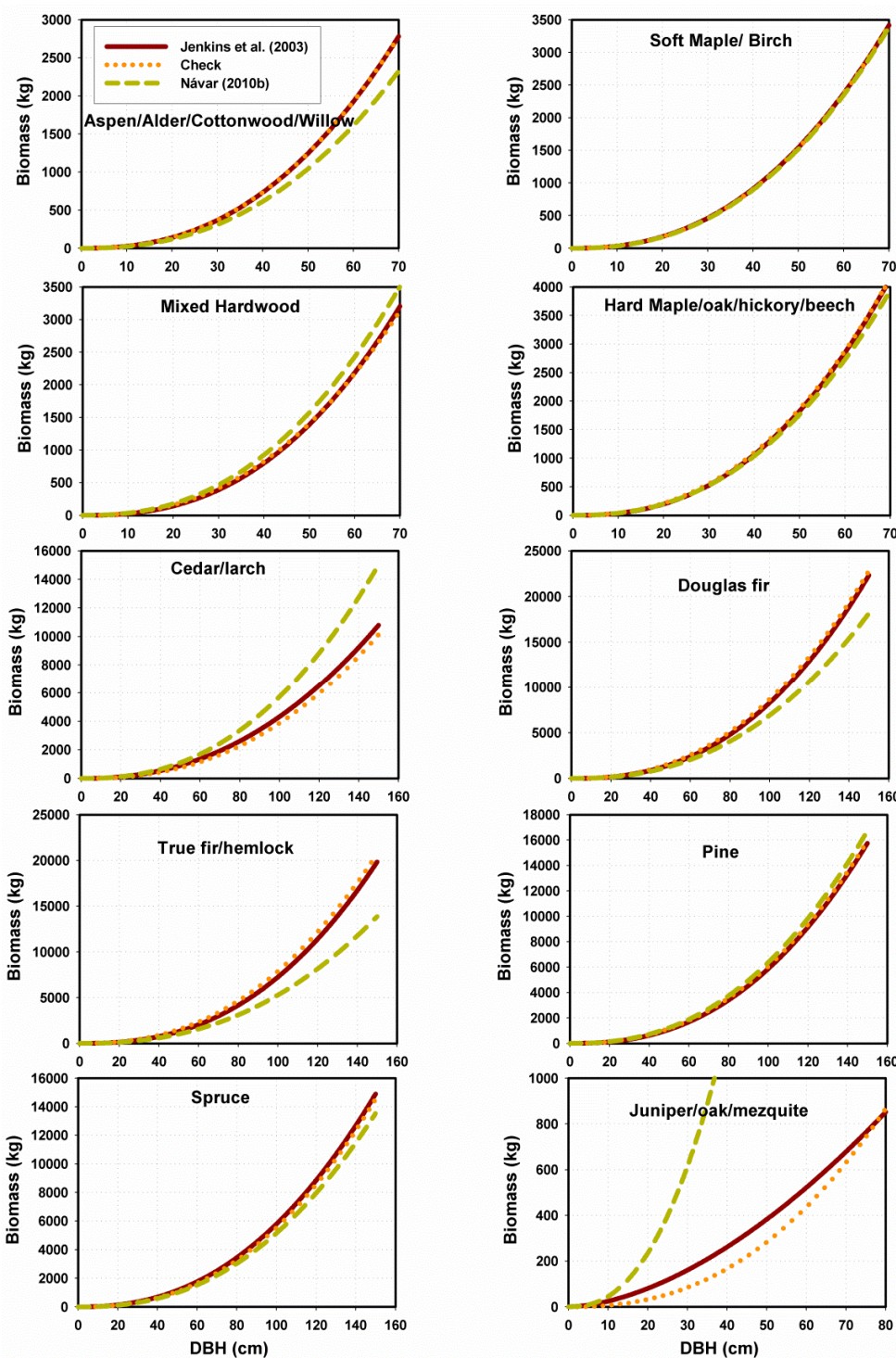

**Figure 4.** The proposed semi-empirical non-destructive restrictive model for tree aboveground biomass estimation contrasting the Jenkins et al. [6,23] equations for 10 clusters of North American tree species.

### 3.2. The $M_{SD}$ Model

### 3.2.1. Preliminary Parameter Assessments and Relationships

The mean (confidence interval) *a* (0.068 ± 0.03 and 0.99 ± 0.03) and *B*-scalar (2.53 ± 0.09 and 2.38 ± 0.16) coefficient values for both datasets are reported in Table 2. The mean (confidence bounds) of the $H^*$ scalar coefficient derived from the relationship $H = aD^{B^*}$ was 0.54 (0.067) for all 21 as well as for all 11 allometric data sets used for fitting and validation

of the $M_{SD}$ model coefficients and parameters (Table 3). The results are significantly similar and within the value range proposed by Niklas [38], who reported a grand average value of 0.535 for a wide range of plant sizes. The mean (confidence interval) values for the $d$ and $h$ coefficients derived from the Schumacher and Hall [34] bole volume equation ($Ln(V) = Ln(a) + dD + hH$; $V = a_v D^d H^h$), were $d = 1.933$ (0.066) and $h = 0.917$ (0.079), respectively. As expected, these parameters fall within the range of $2 < d + h < 3 = 2 < 2.85 < 3$; with lower and upper confidence values of 2.70 and 2.99, respectively. As expected, they approximate better to the 3rd than to the 2nd dimensional domain.

Data on the $B_{empiric}$ and the $H^*$ coefficients for all of the 21 case studies (Tables 2 and 3) were plotted and a statistical relationship was fitted. As expected, the $B_{empiric}$ values fitted well a linear function with the $H^*$ slope of the logarithmic $H$-$D$ relation. However, an exponential rise to a maximum function described the $B_{empiric}$ data better. If a linear equation had been fitted, the equation would have turned out in the following statistics: $B_{empiric} = 2.0099 + 0.9334H^*$; $r^2 = 0.42$; $Sx = 0.16$; $F = 13.85$; and $p = 0.0014$. The F test for selecting the best regression equation did not have significant differences between equations ($p = 0.68$). Integrating the mathematical function solves for the mean and confidence interval parameters of the $B_{empiric}$ values ($2.44 \pm 0.08$). Zianis and Mencuccini [3] developed a similar linear relationship between these parameters, with the following equation: $B_{empiric} = 1.9262 + 0.6972H^*$; $r^2 = 0.42$. The linear or the non-linear equation is of little predictive value for the $B$–scalar coefficient because of the low coefficient of determination and consequently the large unexplained variance. However, the regression is statistically significant pointing to the physical association of the tree $M$-$D$ and the $H$-$D$ relationships. For example, trees trying hard to reach dominant position in an overstocked forest would have large $H$ and small $D$ values with $H^*$ figures $> 0.50$ and $B$-scaling exponents $\approx 2.33$ of the $M$ vs. $D$ relationship. On the other side, wolf trees growing free of competition would be shorter and fat and would have $H^* < 0.50$ and $B$-scaling exponents $\approx 2.67$.

Testing the Association of Equation Parameters

The statistical parameters for the scalar $a$ and $B$ coefficients of Equation (1) for the 21 equations for fitting parameters and the 11 equations for testing the model's performance are reported in Table 3. The $B_{empiric}$ ($2.44 \pm 0.08$) and $B_{theoric}$ ($2.53 \pm 0.09$) parameters reported in Tables 2 and 3 are statistically similar ($p = 0.36$) and linearly and positively related by: $B_{empiric} = 1.0276 + 0.6676B_{theoric}$; $r^2 = 0.42$; $Sx = 0.16$; and $p = 0.0015$. This equation has a large unexplained variation as well given by the small $r^2$ and the large $Sx$ parameters. However, the statistical significant association of the $M_{SD}$ model proposal shows how several physical and biological features converge into the evaluation of tree $M$.

The $a$-Scalar Intercept

The $a$-scalar intercept recorded average values of 0.068 and 0.099 for fitting and testing case study datasets, respectively. As expected, statistical significant associations were found between the $a$-scalar intercept and $a_h$ ($a = 2.035 + 0.43a_h$; $r^2 = 0.45$, $p = 0.0001$); between the $a$-scalar intercept and $a_v$ ($a = -9.05 + 0216a_v$; $r^2 = 0.30$, $p = 0.0001$); as well as between the $a$-scalar intercept and $a_h a_v$ ($a = -18.84 - 3.86\ a_h a_v$; $r^2 = 0.42$, $p = 0.0001$) testing correct the physics of the $M_{SD}$ model describe well the biological traits of M.

A significant negative power relationship between $a_{empiric}$ and $B_{theoric}$ as well as between $a_{empiric}$ and $B_{empiric}$ fitted quite well the data set for all 21 tree M case studies (Figure 5). For the fitting parameter dataset, both equations used to estimate the intercept values for $a_{empiric}$ and the $a_{theoric}$ values converge for the B values larger than 2.2, and this finding encourages the use of empirical equations to estimate $a_{theoric}$ with the following equation; $a_{theoric} = 159.85B_{theoric}^{-8.3852}$. This equation explains 95% of the $a_{theoric}$ variation and has a small standard error of 0.022, with a coefficient of variation of 20%. Other previously reported equations relating a to B are: $a = 38.78B^{-6.84}$; $a = 61.62B^{-7.32}$; $a = 7.03B^{-4.63}$; $a = 5.14B^{-4.49}$; $a = \exp^{(6.98 - 3.71B)}$; $a = \exp^{(3.42 - 2.27B)}$ [3,5,14]. These equations show a great deal of variation in estimating a when B < 2.2 but the Eq. of Zianis and Mencuccini [3] and

Pilli et al. [5] converge quite consistently for the full data range measured in this study. By eliminating one of the Eq. reported by Pilli et al. [5] and using the rest of equations, the a-scalar coefficient has a mean value of 0.11 with a quite small confidence interval of 0.008 when B = 2.40. This figure is quite consistent with the a-scalar intercept values of the meta-analysis conducted in Table 1.

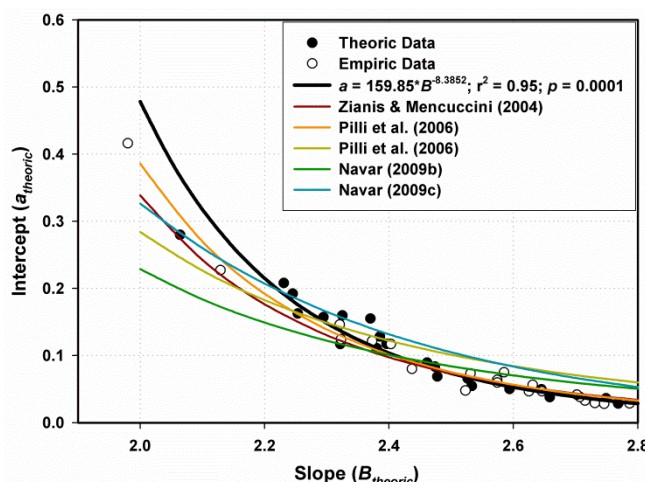

**Figure 5.** Relations of a and *B*-scalar coefficient values reported for 21 case studies (see Table 3). The regression line, equation and parameters are also depicted in conjunction with a set of previously developed equations [3,5,14,16].

The consistency of the empirical data, fitted equation, and other equations previously developed point to the similitude of the relationships for most M case studies. This finding may shed further light into the common development of M for different tree species, better modeling approaches, as well as on the ontogeny of *M* allometry. As trees grow the *B*-scalar exponent of the tree *M* vs. *D* relationship increases non-linearly and the *a*-scalar intercept decreases non-linearly approaching the limits of safe loading mass as the trees grows in height.

### 3.2.2. Performance of Fitting and Validation of the $M_{SD}$ Model

The tested $M_{SD}$ model was: $M_{SD} = (a_{theoric} = D^{Btheoric}) = ((159.85B_{theoric}^{-8.3852})D^{d+h*H*})$. For the 21 *M* case studies, the mean standard error (confidence interval) had a value of 39.8 (17.9) Kg, and it slightly increased to a mean of 47.2 (27.0) kg stressing the consistency of the proposed non-destructive technique in *M* assessments. For six out of the 21 case studies, the standard error calculated by the $M_C$ model was larger than the standard error calculated by the $M_{SD}$ model. The inverse was correct for 15 out of 21 studies. This randomness in *M* estimates from the $M_{SD}$ model, in contrast to the $M_C$ conventional one, makes this methodology unbiased for these *M* case studies (Figure 6). Equation (1) provides excellent *M* assessments for data samples as well as for large datasets but most data samples does not usually meet the sampling requirements.

The $M_{SD}$ model predicts consistent *M* assessments when contrasted with the $M_C$ model, with average r² and Sx% values of 0.96 and 15%, respectively for fitting parameters and 0.96 and 14% for validation of the model (Figures 6 and 7). The $M_{SD}$ model represented by the dark green line oscillates up and down and close to the average dotted line depicted by the $M_C$ model for both datasets pointing to the unbiasedness of *M* assessments by this novel technique.

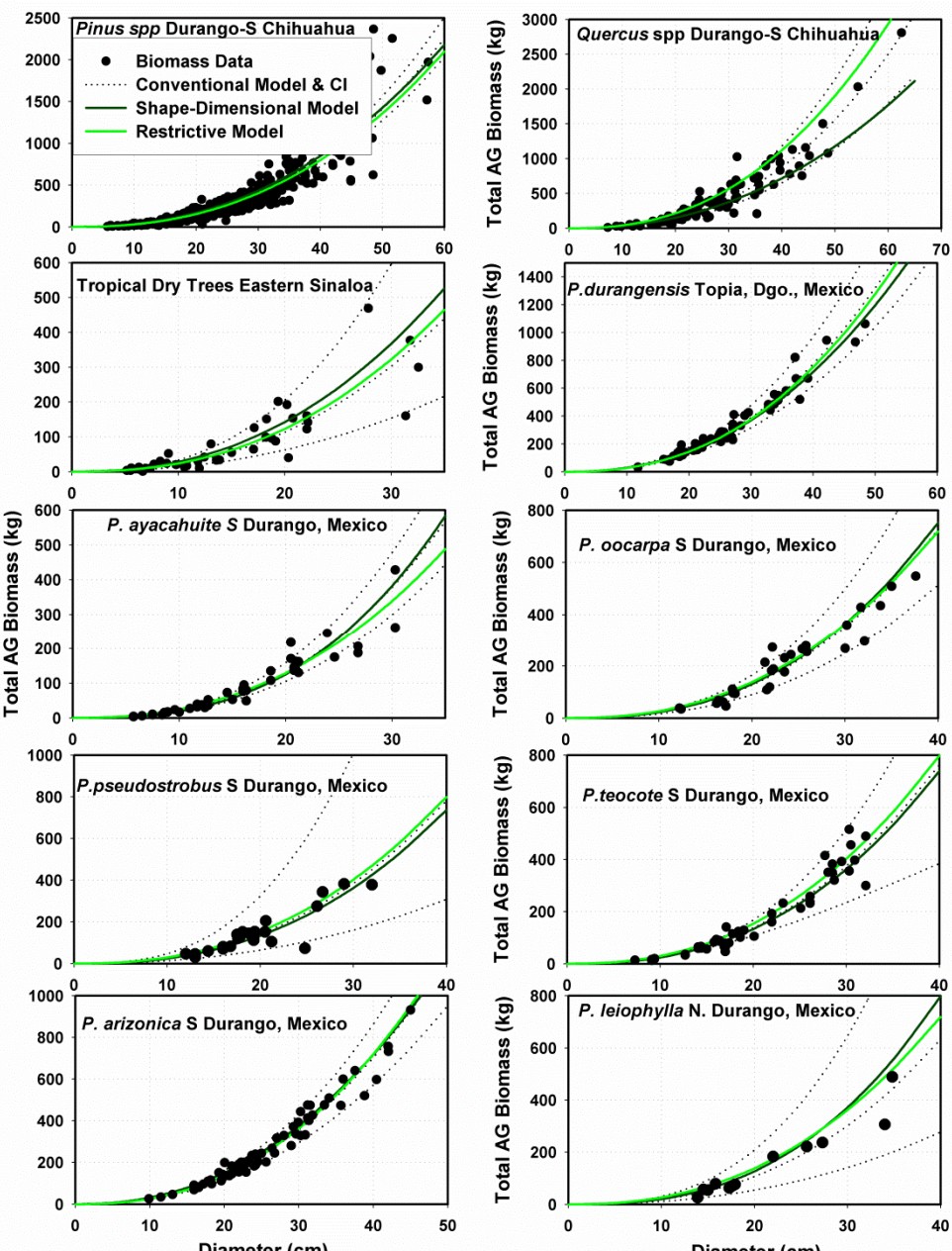

**Figure 6.** Testing the performance of two non-destructive models for 10 selected allometric case studies that were used for fitting parameters. The regression lines, raw data and confidence bands on the empirical *B*-scalar value are also depicted.

Figure 7 shows the independent data set of conventional allometric equations that validated the $M_{SD}$ model performance. In general, the $M_{SD}$ model validated *M* data as well. For nine out of 11 data sets, the model predicted tree *M* within the confidence bounds given by the standard error of *B*. The model did not perform well for *Acacia* spp. neither for *Prosopis* spp. trees with *D* > 30 and *D* > 20 cm, respectively. The bole volume equation used in this research includes branches because the correct definition of the bole component for woodland broadleaf and semi-arid trees is not well specified, and this is one likely explanation for the lack of fit of the $M_{SD}$ model for this type of *M* case datasets.

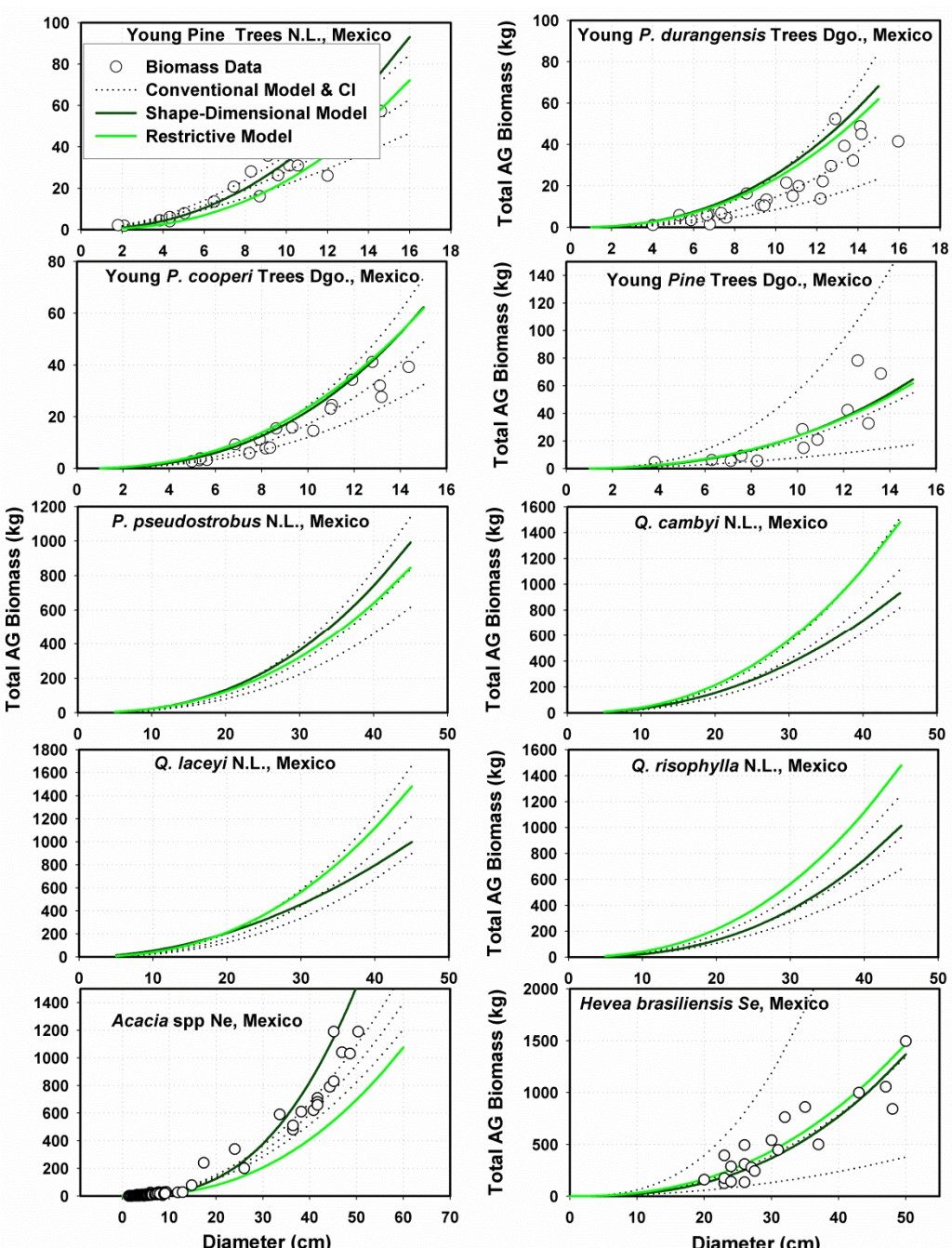

**Figure 7.** Examples for 10 selected allometric case studies that were used for validation of the shape-dimensional and the restrictive model. The regression lines, raw data and confidence bands on the empirical *B* value are also depicted.

### 3.3. Contrasting the Performance of $M_{SD}$ and $M_{NR}$ Models

The $M_{NR}$ model slightly improved $M$ assessments in contrast to the $M_{SD}$ model as the average values of 0.99 and 12% as well as 0.97 and 9% were calculated when contrasting it with the $M_C$ equation for the $r^2$ and the Sx% values for both datasets fitting and validation of the models, respectively. Better approximations between the $M_{NR}$ and the $M_C$ in contrast to between the $M_{SD}$ and the $M_C$ models appear to be related to better quantifications and parameterization of the *a*-and *B*-scalar coefficient of the $M_{NR}$ model.

Contrasts between the $M_{NR}$ vs. the $M_{SD}$ model predictions showed $M$ assessments slightly deviated a bit more than when contrasting them individually with the $M_C$ equation; as average values of 0.97 and 20% as well as 0.65 and 60% were calculated as $r^2$ and Sx%

values for fitting and validation of the proposed models, respectively. Largest deviations occurred in woodland Oak and Acacia tree species. Both proposed non-destructive models performed as good as the $M_C$ model for pine tree species stressing the better parameterization of proposed semi-empirical models in conifers than in woodland broadleaf and semi-arid tree species.

The absolute difference between $a_{empiric}$ and $a_{theoric}$ explained most variation of the absolute difference between $M_{NR}$ and $M_{SD}$ with the following equation; $|M_{NR}$-$M_{SD}|$ = 481.2 + 102.9*Ln($a_{NR}$-$a_{SD}$); $r^2$ = 0.74; $p$ = 0.0001. A preliminary analysis failed to identify any relation between the magnitude of $|M_{NR}$-$M_{SD}|$ and other specific features for each case study (number of sampled trees, $D$ range, and the mean square error for the $D$-$H$ or the $V$-$D$, $H$ relations). The number of trees in each biomass study was statistically related to parameters $a_{empiric}$, $a_{theoric}$, $B_{empiric}$, $B_{theoric}$, Sx1 and Sx2, but the variation explained for by this parameter hardly exceeds 20% of the total variance. Unfortunately, the 21 datasets reviewed do not provide the information that is essential to develop testable hypotheses for disentangling the observed variance of any variability criterion and further analysis did not take place.

## 4. Discussion

### 4.1. The Performance of Semi-Empirical Models

The proposed non-destructive semi-empirical $M_{NR}$ and $M_{SD}$ models perform as good as the conventional empirical, $M_C$, equation when predicting tree $M$ since forecasts lie close and remain within the confidence bounds of the $M_C$ equation. Unexplained variation is smaller than the standard error of most empirical allometric equations reported in the scientific literature (e.g., Sx < 20%). Mean intrinsic deviation is > 15 % of the mean $M$ [6,11,16].

The $M_{NR}$ model has at this time the major advantage that the relationship between $a_r$ and $\rho_w$ provide a wide range of $M$-$D$ relationships that appear they are the case for temperate and boreal tree species. However, the model violates the major assumption tested correct in this study that both $a$ and $\beta$-scalar coefficients varies among tree species and covary together as the tree grows over time. Hence, it hints at a more variable $a$ and less variable $\beta$-scalar coefficient values, at least for temperate and boreal forests. Figure 1a supports this statement as well as the average (and small confidence bounds) of the $\beta$-scalar coefficient of 2.38 ($\pm$0.06) of meta-analysis case studies (Table 1). Although this comprehensive dataset shows also an average (with small confidence bounds) of the $a$-intercept value of 0.14 ($\pm$0.03); the ratio of CB$_\beta$/$\beta$ = 0.021 and CB$a$/$a$ = 0.21 emphasizing the larger variance of the $a$-intercept value.

The $M_{SD}$ model provide consistent $M$ assessments in contrast to the $M_{NR}$ and $M_C$ models as the statistics showed similar $M$ values for all tested case studies. However, the relationship between $a_r$ vs. $B$ provides limited application in tree species with quite small or quite large wood basic density values where both the $a_r$ vs. $B$-scalar coefficients are in the lower or upper range of Figure 5 and predicted scalar coefficients with the negative power equation would deviate $M$ assessments. Hence, at this time the model predict well $M$ but for tree species within the central range of values of $\beta$ = 2.40 and $a$ = 0.10 in tree species with $\rho$ values in the range of 0.35 to 0.70. Using Paz-Pellat et al. [21] scalar coefficients calculated with mathematical equations predict also consistent $M$ assessments but exclusively in the central range of the 39 equations of Ter Mikaelian and Korzukhin [15] and tend to deviate more notoriously than the $M_{SD}$ model outside the likely standard deviation values of these coefficients. Therefore, the mathematical as the $M_{SD}$ and Paz-Pellat et al. [21] models require of better parameterization and the improvement of the required relationships, in specific the $V$ vs. $H$, $D$ and the $H$ vs. $D$ functions That is, these models have to become more physically-based with biological interpretations as an aid to improve mathematical models or the hybrid, physic-statistical version one tested in this research. Examples of how the $M_{SD}$ and Paz-Pellat models perform when predicting biomass for 39 North American temperate and boreal tree species are reported in Figure 8.

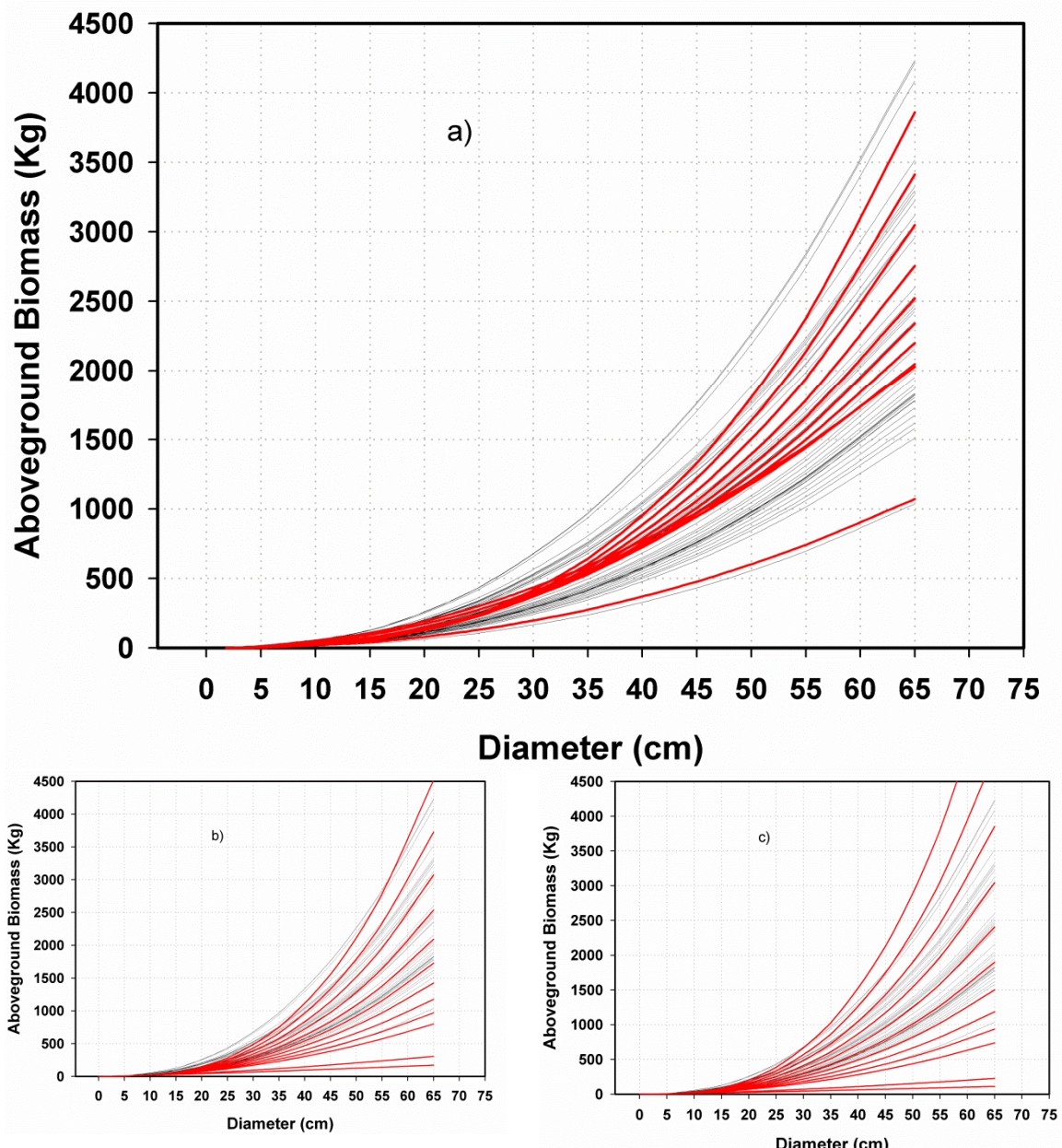

**Figure 8.** Testing the performance of the (**a**) shape dimensional model proposed in this report as well as (**b**,**c**) two forms of Paz-Pellat. [21] models, respectively. The last two models assume the $\beta$-scalar coefficient has been correctly estimated. Red lines represent the models and grey lines the aboveground biomass equations reported by Ter Mikaelian and Korzukhin [15].

### 4.2. Major Assumptions of Semi-Empirical Models

The $M_{NR}$ and $M_{SD}$ models are independent of each other and parameterized in a different manner as well. In addition, they contrast each other in major assumptions of evaluating the scalar coefficients. The $M_{NR}$ model assumes a single constant $B$-scalar exponent value and a $C$ proportionality constant. The $M_{SD}$ model entails variable $a$ and $B$-scalar values. The question whether tree $M$ allometry needs a constant versus a variable $B$-scalar exponent appears to be answered by both models of this report. This research showed that bio-physics point towards a varying $a$ and $B$-scalar coefficients but with a small variance as it was demonstrated in the physical model development as well as in the variation of the statistics reported in the meta-analysis of world case studies.

### 4.2.1. A Constant *B*-Scalar Exponent

The proposed constant *B*-scalar exponent violates the major ontogenetic principles developed in here and supported by Genet et al. [22] findings, However, the constant value assumed in this study and found in this meta-analysis case studies of *M* allometry (2.38 ± 0.06) is an average with a small variance that is consistent with independent estimates using the bio-physical approach. Most meta-analysis *M* case studies should have *B*-scalar values close to 2.40; e.g., Návar [6] for 34 Mexican tree species found a mean value of 2.42 (±0.08) as it was found in the $M_{SD}$ model as well. However, further bio-physical research is required on this approach as well as on any other independent methods such as those conducted earlier [2,11] to be contrasted with the constant value proposed in this research as well as by the $M_{WBE}$ model. The value of *B* = 2.38 (± 0.06) lies in the lower range of ideal geometric objects of tree boles, approaching conical shapes. The $B_{WBE}$-scaling exponent of the $M_{WBE}$ model appears to require refinement, since empirical and physical coefficient values range from 2.32 to 2.44 for 68% and from 2.26 to 2.50 for 95% of the comprehensive allometric equations reported in here, assuming it is normally distributed. Enquist et al. [19] and West et al. [20] noted that the fractal approach was derived assuming the relationship between *D* and *H* scales with an assumed exponent of 2/3. This value is a bit larger than those reported [38] as well as the one measured in this research. Therefore, it is more likely that *D* and *H* scale with an exponent close to ½ instead of 2/3 as it had been more physically explained [35,37], which could help to tune the $B_{WBE}$-scalar coefficient a bit further. The constant *B*-scalar coefficient works fine for World conifer tree *M* equations as long as forests are maintained with the similar dimensions within rotations but they must be on site calibrated whenever it is possible following the bio-physical recommendations proposed in this research.

### 4.2.2. The Variable *B*-Scalar Exponent

Ontogenetic principles developed in this report support the notion the *β*-scalar coefficient varies with age and consequently with the shape of tree boles. In fact, the *B*-scalar exponent value varies for collected World tree species. The $M_{SD}$ model describes well how both the *a*- and *B*-scalar coefficients covary together in most meta-analysis case studies conducted and described well in Figure 5. In addition, a meta-analysis study composed of 279 *M* equations for World tree species, the *B*-scalar coefficient has a mean, a variance and a standard deviation value of 2.3699, 0.077, and 0.278, respectively. The normal and Weibull probability density functions fitted to this data set are displayed in Figure 9.

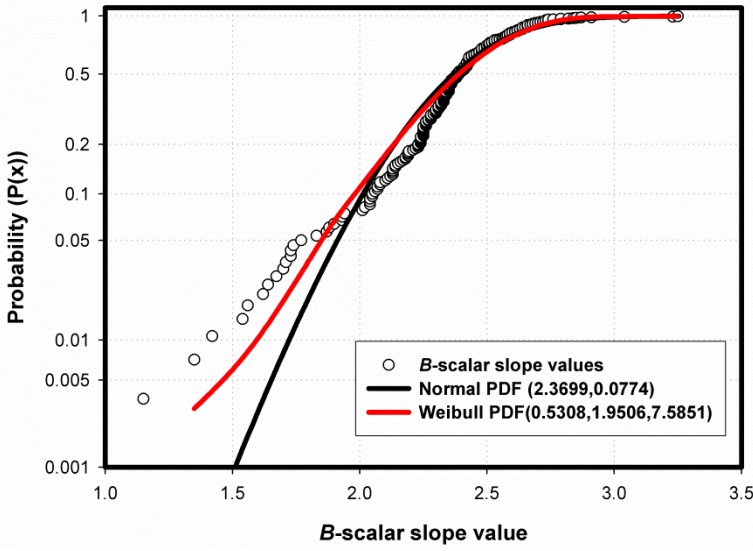

**Figure 9.** The *B*-scalar coefficient value of 279 tree aboveground biomass allometry case studies reported for tree world species (Data Source: Zianis and Mencussini [3]).

### 4.3. Ontogeneity

The variable *B*-scalar exponent figure would violate the main assumption of the $M_{WBE}$ and $M_{NR}$ model proposed in here although a variable *B*-scalar exponent value appears, however, to be the right bio-physical solution because it is consistent with the ontogeny-dependency of *M* proposed earlier [22] and bio-physically described by the $M_{SD}$ model. The ontogeny concept can be further statistically described below as follows. Using the *M* case studies reported [3], the calculated *C* coefficient value is partially dependent upon the *B*-scalar slope parameter as both coefficients fit well with a positive power relationship (e.g., $C = 0.0619*B^{1.6435}$; $r^2 = 0.39$; n = 177; $p = 0.001$). It is also a function of the maximum recorded diameter, $D_m$, since they also fit well with a positive power relationship (e.g., $C = 0.0828D_m^{0.3162}$; $r^2 = 0.24$; n = 177; $p = 0.001$). Ontogeny is then demonstrated because *D* or $D_m$ as a function of tree age fit well with one of the classical sigmoid equations [47], as described using the Richards-Chapman model as in Equations (21)–(23), as follows;

$$D_m = K\left(1 \pm exp^{-bA}\right)^d \tag{21}$$

$$C = 0.0828D_m^{0.3162} \tag{22}$$

$$C = 0.0828\left(K(1 \pm exp^{-bA})^d\right)^{0.3162} \tag{23}$$

where *K, b, d* is the Chapman-Richards sigmoidal equation parameters for the diameter growth equation; *A* is the tree age; *Dm* is the maximum diameter; *C* is the proportionality constant; and 0.0828, 0.3162 are the statistical coefficients.

Recalling from Equation (6);

$$a_r = f\left(\frac{aD_m^B}{D_m^{2.38}}\right); \ a_r = f\left(aD^{B-2.38}\right); \ a = f\left(\frac{a_r}{D^{B-2.38}}\right) \tag{24}$$

As well as from Equation (20):

$$a_{theoric} = f((a_v a_h)\rho); \ a_{theoric} = f(C\rho); \ C = f(a_v a_h) \tag{25}$$

And finally from the following relationship from Figure 5;

$$a = f(B); \ a = f\left(a_{pe}B^{-B_{pe}}\right) \tag{26}$$

All equations from (21) to (25) support further the assumption the *B*-scalar slope varies as a function of the main tree bole traits pointing to the need of tuning the fractal Equation (2) as well as the $M_{NR}$ models. Tree bole geometry from *D* to *H* or from *Db* to *H* shift from quasi-conical to quasi-cylindrical shapes, as a function of several factors, as a consequence either *V* or *M* shift scalar coefficients as in Equation (26) for conical and Equation (27) for cylindrical shapes:

$$V = \frac{1}{3}0.7854(D^2H); \ M = \rho\frac{1}{3}0.7854(D^2H) \tag{27}$$

$$V = 0.7854(D^2H); \ M = \rho0.7854(D^2H) \tag{28}$$

### 4.4. Improving Scalar Parameter Assessments

#### 4.4.1. Scalar Coefficients

The extreme values of the scalar coefficients; $2.33 \leq B \leq 2.67$; $0.08 \leq a \leq 0.26$; $0.16 \leq C \leq 0.52$ of Equations (3)–(5); in addition to the uncertainty assumed by $\rho_w = \rho$ as well as the covarying relationship $a = a_{pe}B^{-Bpe}$ point to the need of improving parameter assessments using bio-physical principles as the most accurate solution to evaluate *M* more precisely for any forest as at this time the large variance hinders the development

of semi-empirical and theoretical models. The $a$-scalar intercept and the $C$ proportionality constant fall within the averages and confidence bounds of these extreme values for several temperate tree species [6]. The assumption of a fixed $B$-scalar coefficients and a variable $a$-scalar intercept provides excellent preliminary results as demonstrated by the $M_{NR}$ model but needs further bio-physical interpretation and describe better its potential variability to be consistent with the $M_{SD}$ model. In doing so, several preliminary approaches can be exhausted: (i) parameterize more precisely the $C$ proportionality constant value using the mathematical solution of Equation (15), the multiplication of Equation (17) times Equation (18), develop better relationships between $a$ and $\rho$, among others; (ii) using statistics parameterize more precisely the $a$-scalar intercept value with the use of: more extensive data sets, re-escalate with Equation (7) as Figure 1 provides evidence how the variable $a$-scalar intercept solves the $M$ variability, using the $M_{SD}$ model, develop more precise relationships between $V = f(D,H)$ e.g., fitting the Schumacher and Hall [34] equation as well as the $H = f(D)$ to evaluate more precisely $a_v$ and $a_h$, develop together geometric and mathematic principles for tree boles, among others; (iii) test the assumption $\rho_w = \rho$ or weight accordingly by measuring these components separately; (iv) extract the variance of each parameter to establish proper confidence bounds. Several conclusions can be preliminary drawn: (i) the $a$-scalar intercept is a complex coefficient that is a function of bole shape and wood basic density; (ii) the statistical relationships fitted in this report between $a$ vs. $a_v$, $a$ vs. $a_h$, $a$ vs. $a_v \cdot a_v$ as well as the bio-physical principles of the $M_{SD}$ model reinforce the notion of the dependency of $a$ upon the shape of tree boles as almost half of the variation is explained by the shape coefficients and as described by Figure 2 the other half is explained by the wood basic density parameter. The multiplication of Equation (17) times Equation (18) for 80 pine trees of Central Durango Mexico result in an average (standard deviation) $C$ value of 0.1932 with 95% confidence bunds of 0.179 and 0.207 pointing that the $C$ coefficient value differs from the 0.2457 value with 95% confidence bounds of 0.23 and 0.26 for North American temperate and boreal conifers of Figure 2. Note that both techniques of evaluating the $C$ proportionality constant are different as well as that they are used for two different tree communities. Both techniques of evaluating $C$ should be applied to both tree communities to draw better conclusions on better and simple ways of assessing the $a$-scalar intercept value. The $B$-scalar coefficient can be mathematically defined as the $M$ growth rate per unit change in $D$ and the power coefficient describes well the acceleration rate.

However, Figure 1 shows that variable $B$-scalar coefficient would deviate more from measurements or conventional empirical relationships than with changes in the $a$-scalar intercept value pointing to changes in the shape of tree boles have a better control than the growth rate of the relationship $M$ vs. $D$. In this regard it is important to carry a sensitivity analysis the scalar coefficients have on $M$ for real tree populations or communities with scalar coefficients evaluated with different techniques.

It is clear then that any other tree variable (height, volume and mass) that is associated with $D$ has to be also linked to tree age; e.g., trees with the same $D$ most likely have different $H$, $V$, $M$ values that depends on $C$: the slenderness, $a_h$, and form, $a_v$, factors. Therefore, age-related as well as forest management variables controlling the $C$ shape parameter could be stand stocking or density, tree diversity, ecological relationships between neighboring trees, stand productivity, among others. Consistent with the $M_{NR}$ model assumptions, as well as with the bio-physical principles of the $M_{SD}$ model, Genet et al. [22] demonstrated the $\beta$-scalar coefficients of allometric equations fitted to $M$ vary as a function of age for *Fagus* plantations of Northern Europe.

The fact $M_{NR}$ and $M_{SD}$ models fail to portray precise $M$ assessments for the Jenkins et al. [23] cluster of woodland tree species junipers, mesquite and dry-oak species as well as for cedar/larch and true fir/hemlock trees with $D > 60$ cm provides insights into the potential variation of the $C$ parameter. Its mean value found in this meta-analysis case study was 0.2457 (0.0152) and should be replaced by 0.025, 0.098, and 0.18 for the groups of species reported above, respectively. With these new coefficient values the proposed $M_{NR}$ model would match the empirical equation $M$

assessments with Sx < 5.0%. The small $C$-coefficient value (0.025) appears to be unlikely found in future formulations and it is likely that a modification of both the $B$-scalar slope and the $C$ coefficient must improve $M$ evaluations in woodland tree species. This observation points out to the need for finding the right $C$ values for the tree species of interest. Pilli et al. [5] showed, consistent with this research, the relationship $a = f(\rho_w)$ must be worked out separately for different forest maturity stages since they fitted two exponential equations ($Ln(a) = -3.12 + 1.11\rho_w$; and $Ln(a) = -3.51 + 1.27\rho_w$); the first one for mature and the second one for adult tree species. Further exploration of these mathematical functions suggests they do not significantly deviate from a linear relationship ($p = 0.37$) with a common, joint equation of: $a = 0.1264\ (\pm 0.0089)\rho_w$ for a $\rho_w$ range of 0.20 to 0.90. Návar [6] noted the $C$ coefficient value varies from 0.17 to 0.31 for the clusters of species reported by Jenkins et al. [23]. Therefore, the $C$ proportionality coefficient must be better understood for dryland forests and calculated for at least each individual dominant tree species.

### 4.4.2. The Wood Specific Gravity Value

An understanding and better descriptions of the $a_r = f(\rho_w)$ relationship leads to the conclusion that there is some room for improving this relationship and consequently to better assess tree $M$ but first other sources of $\rho_w$ error must be understood and evaluated. Wood specific gravity varies within the same tree [48] and between trees growing in similar or different environmental conditions [4,11,48]. Within an individual tree, $\rho_w$ changes from the tree bottom to the top and from the pith to the bark [48] as well as among $M$ components (bark, hardwood, softwood, foliage, branches) [6]. Therefore, conventional standardized methods of measuring $\rho_w$ must be re-evaluated by considering these variations. For example, [49] come up with guidelines to collect wood cores at $0.22H$ when they are conventionally taken at fixed $D$; $H = 1.30$ m. Further research is also required on the mathematical function that describes the relationship between $\rho_w$-$D$ to complete this mathematical task and finally matching $\rho_w$ and $\rho$. Note that three functions must be weighted: (i) bark, (ii) softwood, and (iii) hardwood that compose $D$ at any tree bole $H$. This complicates any mathematical development of the $\rho_w$-$D$ relationship. Miles and Smith [50] reported valuable information on bark and wood specific gravity values for North American tree species that can aid in the development of the $\rho_w$-$D$ relationship but information on how $\rho_w$ changes in the softwood and hardwood bole components should be at least evaluated in order to use weighting factors. In addition, $\rho_w$ varies greatly as a function of altitude and geographic regions stressing the need for measuring this parameter in conventional allometric biomass studies [4]. Simple, easy, flexible, and portable techniques are required to measure this variable in the field and to be modeled in the laboratory. In the meantime, several lists of $\rho_w$ for the most important commercial North American [23,50] and for several tropical [1,4] tree species are now available.

### 4.4.3. Proper $a$- and $B$-Scalar Coefficient Values

Should tree bole volume, $V$, be characterized by exponent values ($2.0 < V < 3.0$), $M$ must also be featured by exponent values $2.0 < M < 3.0$. The volume for tree boles with conical, $V = 0.2618(D^{2.0}H^{1.0})$, or cylindrical, $V = 0.7854(D^{2.0}H^{1.0})$ shapes must be escalated to tree $M = \rho(0.2618(D^{2.0}H^{1.0}))$ for conical or $M = \rho(0.7854(D^{2.0}H^{1.0}))$ for cylindrical shapes. Tree boles have a wood specific gravity value of less than $1.0$ g cm$^{-3}$ for most tree species and conifer trees have an average value of $\rho \approx 0.50$ g cm$^{-3}$; then $M \approx (0.1309(D^{2.0}H^{1.0}))$ for conical or $M \approx (0.3927(D^{2.0}H^{1.0}))$ for cylindrical shapes. Most average coefficients of the Schumacher and Hall's bole volume equation; $D^d$ and $H^h$ are smaller than $d = 2.0$ and $h = 1.0$. In fact, average (confidence interval) values found in this study were: $d = (1.93 \pm 0.06)$ and $h = (0.92 \pm 0.07)$ and so are the $a_v = 0.1309$ and $a_v = 0.3927$ values of tree boles with conical and cylindrical shapes for conifer trees. The addition of branches and foliage into the aboveground biomass component of forest ecosystems would slightly lower even further these coefficient values. Moreover, the simplification of the explanatory variable; e.g., $M = f(D)$, implies $M = \rho_w(a_v a_h D^{d+hH^*})$; as $a_h$ and $H^* \approx \frac{1}{2}$; then: $0.01 < (a_v + a_h) = C\rho_w < 0.20$;

and $B \approx 2.0 + 1.0(0.50)$. For conifer trees, any $M$ equation exceeding these coefficient values would violate the bio-physical principles of tree allometry that would probably reflect most likely problems with understanding and applying sampling theory. Simulations conducted on the scaling coefficients that randomly vary from the population means when taking biomass datasets with different sample sizes tested how sample size plays key roles in defining the population coefficient value [6].

### 4.5. Remarks

A final note on the $M_{NR}$ and $M_{SD}$ models proposed in here is that they can also be employed to double check the consistency of reported conventional allometric equations. That is, whether equations that crosses the $a_r$-intercept lines would bias $M$ assessments. The limits of most empirical allometric equations can be easily determined using this non-destructive approach. In other words, the biomass equation limits of application can be found just before they cross the $a_r$ lines. Hence, this technique is handy for finding the right equations and their limits of application in the field, and as a consequence $M_{NR}$ or $M_{SD}$ can be used to improve $M$ estimates for any tree species of any forest that lack tree allometry as well.

### 4.6. Summary of Allometric Models to Evaluate Tree Aboveground Biomass

Several semi-empirical models had been developed using different approaches from statistics ($M_C$, $M_{CH}$), fractals ($M_{WBE}$), geometric ($M_{GE}$), and bio-physics ($M_{SD}$) as well as hybrid ($M_{NR}$ & $M_{SD}$) approaches. A summary of these models is reported in Table 4.

**Table 4.** Description of common equations and models reported in the scientific literature for tree aboveground biomass assessments contrasted with the two semi-empirical non-destructive models tested in this report.

| Name | Equation or Model | Parameters | Calculation Method | Spatial Range | Author |
|---|---|---|---|---|---|
| Conventional | $Ln(M) = Ln(a) + BLn(D)$ <br> $M = aD^B$ | $a, B$ | Statistics | Species, Local | Baskerville [17] |
| Fractals ($M_{WBE}$) | $M = C\rho D^{2.67}$ | $C, B = 2.67$ | Statistics, Mathematics | Forests, Worldwide | West et al. [18] |
| Restrictive ($M_{NR}$) | $M = C\rho D^{2.38}$ <br> $a_r = C\rho_w; a_r \neq a$ | $C, B = 2.38$ | Statistics, Mathematics | Forests, Worldwide | This Report |
| Shape-Dimensional ($M_{SD}$) | $M = C\rho D^{d+hH^*}$ <br> $V = a_v D^d H^h; H = a_h D^{dH^*}$ <br> $C = a_v a_h$ | $d,h,H^*,a_v,a_h$ <br> $2.33 \leq B \leq 2.5$ <br> $H^* \approx 0.54$ | Statistics, Mathematics, Physics | Forests, Worldwide | This Report |
| Genet ($M_{GE}$) | $M = a + \rho F(D^2H)^\gamma$ | $a, F, \gamma$ <br> $\gamma \leq 0.95$ | Statistics, Mathematics | Groups of Species | Genet et al. [22] |
| Ketterings ($M_{KE}$) | $M = C\rho D^{2+H^*}$ | $C, H^*$ <br> $H^* \approx 0.53$ <br> $B \approx 2.53$ | Statistics, Mathematics | Tropical Forests | Ketterings et al. [2] |
| Chave ($M_{CH}$) | $M = C(p_w D^2 H)^\gamma$ | $C, \gamma$ <br> $\gamma \leq 0.95$ | Statistics, | Tropical Forests | Chavé et al. [10] |

$B$ and $a$-scalar coefficients are conventionally estimated by least square techniques in regression analysis usually using the logarithmic transformation of $M$ and $D$ data and frequently counting for weighted regression, according to model (1); Note that $M_{SD}$ can also be simplified as: $B_{theoric} = d + hH^*$; $a_{empiric1} = M/D^{Bempiric}$; $a_{theoric} = f(B_{theoric})$; Note that $a_{empiric1}$ is also a mean value derived from each individual $M$-$D$ pair data for each biomass data set. Also note that the Spurr & Barnes [51] volume equation ($V = a(D^2H)^B$) is used in $M_{GE}$ and $M_{CH}$ and the Schumacher & Hall [34] volume equation in $M_{SD}$. Former volume equation uses the notion of a tree bole cylinder shape that is weigthed by simple ($C,a,\gamma$) while the later uses more complex tree bole form factors ($a_v,d,h,H^*$). The evaluation of the $C$ proportionality constant value has been discussed in Navar [6] for North American tree species with an average of 0.2457, 0.19 for pine trees of Central Durango, Mexico, and 0.11 for tropical trees of SE Asia. However, note, as expected $B$-scalar coefficient values for NA tree species has an average value of 2.38 and for tropical tree species of SE Asia 2.53. Had the $M_{KE}$ model used a $B = 2.38$, the $a$-scalar intercept would have been = 0.11 instead of 0.066 and the $C$ value would have almost doubled from 0.11 to 0.18 making both the $M_{NR}$ and the $M_{KE}$ compatible models of tree $M$ assessments.

This summary contrasts and points similitudes and divergences in simplifying $M$ allometry and the specific conclusion that can be drawn is that understanding tree allometry continues to be a major challenge as the form of tree boles and the constant or varying

*B*-scalar coefficient value continues to be a major scientific dispute. Better approximations of *M* had been achieved with the inclusion of tree height, *H*, and wood specific gravity, $\rho_w$, values that make equations and models more generalist by incorporating one trait of tree diversity of most natural forests. More complex form factors are proposed in the $M_{SD}$ model but they will eventually require further simplification to be conventionally used, for example, in experimental or commercial forest inventories. At this time, it appears the Schumacher and Hall [34] bole volume equation approximates better the complex form of tree boles than the typical Spurr and Barnes [51] equation.

## 5. Conclusions

The main objective of this research was to simplify tree *M* assessments by developing, testing, and validating two independent semi-empirical models that are consistent with the empirical as well the theoretical models using comprehensive data sets. These techniques are named the restrictive, $M_{NR}$, and the shape-dimensional, $M_{SD}$, models. Both models provide: (i) consistent tree *M* assessments when contrasted with empirical data or with the conventional allometric equation; (ii) important additional information on tree allometry (e.g., how to bio-physically evaluate scalar (*a*, *B*, *C*) coefficients of tree *M* equations and models; the likely linkages and covarying functions between scalar coefficients; most likely values for several tree species and groups of species; limits of application of developed equations; among others); (iii) key understanding scientific issues of tree *M* allometry, e.g., further supporting the notion of ontogenetic-dependency of coefficients; (iv) linkages between biological and physical model assumptions and parameters; and (v) other important data. These two novel approaches can be further improved, in particular the $M_{SD}$, as it is consistent with ontogenetic principles and provides good assessments but only for tree species that have average $\beta$-scalar coefficients found in most meta-analysis studies. Finding the right local or regional parameters for the tree species or groups of species of interest would improve *M* evaluations. These methodologies can also check on the consistency of most reported or newly-developed allometric biomass equations.

**Author Contributions:** S.C.-R. and J.-G.C. helped to check the manuscript and financing costs of publication, J.E.L.-S., T.G.D.-G., J.J.C.-R. and J.d.J.G.-L. helped with financial assistance, F.d.J.R.-F., J.-G.C. and J.d.J.G.-L. helped to collect part of the field data and financial assistance, J.N. organized the research, data collection, data analysis, model techniques, and writing the report. All authors have read and agreed to the published version of the manuscript.

**Funding:** This research received no external funding. Most of this data was collected using personal funding.

**Data Availability Statement:** J.N. has this data source in his laboratory and data is available upon request.

**Acknowledgments:** The authors of this manuscript wish to thank all reviewers for the constructive criticisms that helped to improve the technical content and English readability. Many people are recognized for the aid in the field data collection.

**Conflicts of Interest:** The authors declare no conflict of interest.

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
