# Peer review of "Semi-Empirical Models and Revision of Predicting Approaches of Tree Aboveground Biomass Assessments"

_forests, doi:10.3390/f13070999_

Round 1

Reviewer 1 Report

Dear Author,  I appreciate to conduct such type of study, however, I feel there is a lot of scope to improve the the study such as much better to write a concise abstract. Introduction has to be rewritten and make a flow to understand it by a common reader. There is too much  unnecessary information is added which can go to appendix or deleted. Format is not uniformly managed across the whole manuscript. 

Author Response

Dear  Reviewer:

Please find attached the response IN CAPITAL LETTER EMBEDDED WITHIN YOUR COMMENTS. I appreciate very much your help on improving the technical content and readability of the manuscript.

Kind Regards

Sincerely

Jose Navar, PhD

Oklahoma State University.

REVIEWER 1

Dear Author,  I appreciate to conduct such type of study, THANKS FOR EMPHISIZING THE TECHNICAL IMPORTANCE OF THIS MANUSCRIPT however, I feel there is a lot of scope to improve the the study such as much better to write a concise abstract. ABSTRACT HAS BEEN EXPANDED TO STRESS THE IMPORTANCE OF THE STUDY. Introduction has to be rewritten and make a flow to understand it by a common reader INTRODUCTION NOW FLOWS BETTER BY ADDING SEVERAL COMPONENTS INCLUDING THE IMPORTANCE AND A FURTHER SCIENTIFIC EVIDENCE OF THE NEED TO CONDUCT SUCH A RESEARCH. There is too much  unnecessary information is added which can go to appendix or deleted SOME INFORMATION WAS DELETED BUT SOME REMAIN TO POINT THE NEED OF SUCH INFORMATION. Format is not uniformly managed across the whole manuscript.  THE STRUCTURE HAS BEEN REARRANGED. IF YOU HAVE SOME SUGGESTIONS PLEASE

Reviewer 2 Report

The extensive study presents two very interesting and promising approaches to estimating tree aboveground biomass with the help of semi-empirical models. The results are presented consequently, both propsed solutions are deeply discussed and I guess that study will be point of interest for many readers of Forestry journal. I also found the whole paper well written with the informative charts and tables.

Author Response

REVIEWER 2

The extensive study presents two very interesting and promising approaches to estimating tree aboveground biomass with the help of semi-empirical models. The results are presented consequently, both propsed solutions are deeply discussed and I guess that study will be point of interest for many readers of Forestry journal. I also found the whole paper well written with the informative charts and tables. THANKS FOR THE CONSTRUCTIVE COMMENTS.

Reviewer 3 Report

This is actually a review or meta-analysis paper but the tile and the expressions in the text are not clearly stated, causing some misunderstandings somehow. I have read the paper several times and found it is a bit tricky to disentangle the review content and the analysis conducted by the authors in the manuscript. I suggest revising the title and streamlining the text to stress the key results and findings. In addition, the language could be improved. In summary, I suggest the authors make a thorough revision on the manuscript by considering following comments.

Major comments:

  1. The merits and shortcomings of the recommended models should be more clearly stated in both the Abstract and conclusions and better discussed in the Discussion section.
  2. The quality of the figures could be significantly improved.
  3. The English language should be improved.
  4. The section and subsection tiles should be revised to better organized the logics of the paper.

Minor comments:

L50: “an important key component”?

L205: Who has proposed?

L206: “The proposed models … could be modeled”?

Fig.3: coding errors exist in the figure.

L465: What do you mean by “finding standard errors”?

L635: “proposed models”? Please use more informative words.

L97: These sections describing the conventional methods should be shortened and moved to the introduction part.

Author Response

Dear  Reviewer:

Please find attached the response IN CAPITAL LETTER EMBEDDED WITHIN YOUR COMMENTS. I appreciate very much your help on improving the technical content and readability of the manuscript.

Kind Regards

Sincerely

Jose Navar, PhD

Oklahoma State University.

REVIEWER 3

This is actually a review or meta-analysis paper THE TITLE HAS BEEN MODIFIED TO DESCRIBE THE REVIEW AS WELL AS THE NEED TO DEVELOP BOTH MODELS. YOU ARE RIGHT DATA REVISION ANALYSIS, FITTING AND TESTING PROCEDURES MAKES IT A REVIEW AND A RESEARCH ARTICLE. but the tile and the expressions in the text are not clearly stated, causing some misunderstandings somehow THE TITLE HAS BEEN MODIFIED, IT READS NOW: SEMI-EMPIRICAL MODELS AND REVISION OF PREDICTING APPROACHES OF TREE ABOVEGROUND BIOMASS ASSESSMENTS. I have read the paper several times and found it is a bit tricky to disentangle the review content and the analysis conducted by the authors in the manuscript. THE RESEARCH ARTICLE USES WORLD DATASETS AS WELL AS TO LOCAL DATASETS AND POINTS THE MODELING TECHNIQUES AVAILABLE IN ORDER TO MAKE THE POINT OF THE NEED TO DEVELOP AND VALIDATE BOTH CONTRASTING TECHNIQUES. I suggest revising the title and streamlining the text to stress the key results and findings. REVISIONS ARE CLEARILY DESCRIBED AND FINDINGS ARE CLEARLY STATED AS BOTH KINDS OF MODELS APPEAR TO PROVIDE ADDITIONAL INFORMATION AS WELL AS GOOD M PREDICTIONS AS A MEANS TO VALIDATE THEM. HOWEVER FOR MORE EXTENSIVE DATASETS, THE Mnr APPEAR TO PREDICT BETTER  THEN THE Msd AGB OR M. In addition, the language could be improved. THE MANUSCRIPT WAS SENT TO A COUPLE OF LANGUAGE EDITING COMPANIES. PLEASE PROVIDE THE PLACES WHERE THE MANUSCRIPT CAN BE IMPROVED IN ENGLISH LANGUAGE. In summary, I suggest the authors make a thorough revision on the manuscript by considering following comments.

Major comments:

  1. The merits and shortcomings of the recommended models should be more clearly stated in both the Abstract and conclusions and better discussed in the Discussion section. A BETTER DISCUSSION WAS ADDED STRESSING HOW IN SPITE OF THE SHORTCOMINGS OF THE Mnr OF HAVING A CONSTANT B-SCALAR COEFFICIENT PROVIDES BETTER PREDICTIONS THAN Msd. THEREFORES IT IS RECOMMENDED TO USE THE Mnr PRELIMINARILY IN THE MEANTIME BETTER APPROXIMATIONS ARE ACCOMPLISHED BY IMPROVED  PARAMETERIZATION PROCEDURES OF Msd.
  2. The quality of the figures could be significantly improved. THE QUALITY OF FIGURES HAS BEEN IMPROVED AND SAVED IN A TIFF FORMAT TO KEEP MAJOR FEATURES INTACT.
  3. The English language should be improved. PLEASE PROVIDE WHERE THE ENGLISH LANGUAGE CAN BE IMPROVED.
  4. The section and subsection tiles should be revised to better organized the logics of the paper. A BETTER ORGANIZATION OF THE SECTIONS AND SUBSECTIONS IS NOW PROPOSED.

Minor comments:

L50: “an important key component”? OK. IT IS AN IMPORTANT PARAMETER. THANKS.

L205: Who has proposed? OK. CHANGING PROPOSED BY THE.

L206: “The proposed models … could be modeled”? OK. INSTEAD OF MODELS, PROPOSALS. YHANKS.

Fig.3: coding errors exist in the figure. I AM NOT SURE WHAT DO YOU MEAN BY CODING ERRORS. THE FIGURE IS CORRECT. THE LABELS AND CAPTION ARE CORRECT.

L465: What do you mean by “finding standard errors”? FINDING HAS BEEN REPLACED BY THE.

L635: “proposed models”? Please use more informative words. PROPOSED HAS BEEN REPLACED BY SEMI-EMPIRICAL.

L97: These sections describing the conventional methods should be shortened and moved to the introduction part. OK. GOOD POINT. BUT THE SECTION IS FUNDAMENTAL TO MAKE THE SEMI-EMPIRICAL MODELS MORE UNDERSTANDABLE AND HOW THEY MEET THE ASSUMPTIONS OF THE EMPIRICAL AND THEORETICAL MODELS. HOWEVER, THE SECTION WAS PARTIALLY MODIFIED AND SHORTENED.

Reviewer 4 Report

Dear Authors,

Please find my comments in the attached file.

Regards

Author Response

Dear  Reviewer:

Please find attached the response IN CAPITAL LETTER EMBEDDED WITHIN YOUR COMMENTS. I appreciate very much your help on improving the technical content and readability of the manuscript.

Kind Regards

Sincerely

Jose Navar, PhD

Oklahoma State University.

REVIEWER 4 ATTACHMENT

Is this a new chapter? NO IT IS NOT.

Please check the style of letters in the table it is not palatino linotype. The size of the letters is also too big. OK. I GUESS I WILL DO THAT WHEN THE FINAL VERSION OF THE MANUSCRIPT IS READY. IN THE MEANTIME I HAVE CHANGED THE SIZE OF LETTERS.

Please check the legend. I dont think so that this type of "chinese" letters are alowed, Ny--ar(2010b). Also the name of tree species usulyis not written on the graphs. Instead a, b, c, d, letters are used and their meaning explained in the caption. SORRY TO DISAGREE. IT IS NOT BETTWE TO HAVE THE NAMES OF TREE SPECIES INSIDE THE FIGURE TO SAVE TIME AND TROUBLES GOING BACK AND FORTH THE CAPTION AND THE FIGURES???

Please check the previous comment. SAME RESPONSE.

The same. Please check the figure 5 comment. SAME RESPONSE.

I think you discussed many aspects of your developed models, discluding the practical ones. Please discuss how of if your proposed models could be used to estimate biomass in other countries as well? I think this would be very interesting aspect worth to take into account.

After the conclusion part also goes aknowledgment and contribution parts. I think they are missing? OK. THANKS. THE MANUSCRIPT HAS NOW THE SECTIONS ACKNOWLEDGMENT AND CONTRIBUTION PARTS.

This is what it missing in the introduction part. You clarify research problem only in conclusion part. THE INTRODUCTION HAS NOW WHAT YOU HIGHLIGHTED. THANKS.

How broad your models could be applied, for example Europe? Clarify the possibilities of your models, speculate little bit. DONE. THANKS. IT READS NOW: These two novel approaches can be further improved, in particular the MSD, as it provides good assessments but only for tree species that have average β-scalar coefficient found in most Meta-analysis studies. Finding the right local or regional parameters for the tree species or groups of species of interest would improve M evaluations.

Round 2

Reviewer 4 Report

Dear Authors,

Thank you for taking my comments into account.

Regards